



# How can expert knowledge increase the realism of conceptual hydrological models? A case study in the Swiss Pre-Alps

Manuel Antonetti[1,2], Massimiliano Zappa[1]

[1] Swiss Federal Research Institute WSL, Birmensdorf, 8903, Switzerland
[2] Department of Geography, University of Zurich, Zurich, 8057, Switzerland

*Correspondence to*: Manuel Antonetti (manuel.antonetti@wsl.ch)

**Abstract.** Both modellers and experimentalists agree that using expert knowledge can improve the realism of conceptual hydrological models. However, their use of expert knowledge differs for each step in the modelling procedure, which involves hydrologically mapping the dominant runoff processes (DRPs) occurring on a given catchment, parameterising these processes within a model, and allocating its parameters. Modellers generally use very simplified mapping approaches, applying their knowledge in constraining the model by defining parameter and process relational rules. In contrast, experimentalists usually prefer to invest all their detailed and qualitative knowledge about processes in obtaining as realistic spatial distribution of DRPs as possible, and in defining narrow value ranges for each model parameter.

Runoff simulations are affected by equifinality and numerous other uncertainty sources, which challenge the assumption that the more expert knowledge is used, the better will be the results obtained. To test to which extent expert knowledge can improve simulation results under uncertainty, we therefore applied a total of 60 modelling chain combinations forced by five rainfall datasets of increasing accuracy to four nested catchments in the Swiss Pre-Alps. These datasets include hourly precipitation data from automatic stations interpolated with Thiessen polygons and with the Inverse Distance Weighting (IDW) method, as well as different spatial aggregations of Combiprecip, a combination between ground measurements and radar quantitative estimations of precipitation. To map the spatial distribution of the DRPs, three mapping approaches with different levels of involvement of expert knowledge were used to derive so-called process maps. Finally, both a typical modellers' top-down setup relying on parameter and process constraints, and an experimentalists' setup based on bottom-up thinking and on field expertise were implemented using a newly developed process-based runoff generation module (RGM-PRO). To quantify the uncertainty originating from forcing data, process maps, model parameterisation, and parameter allocation strategy, an analysis of variance (ANOVA) was performed.

The simulation results showed that: (i) the modelling chains based on the most complex process maps performed slightly better than those based on less expert knowledge; (ii) the bottom-up setup performed better than the top-down one when simulating short-duration events, but similarly to the top-down setup when simulating long-duration events; (iii) the differences in performance arising from the different forcing data were due to compensation effects; and (iv) the bottom-up setup can help identify uncertainty sources, but is prone to overconfidence problems, whereas the top-down setup seems to accommodate uncertainties in the input data best. Overall, modellers' and experimentalists' concept of "model realism"



differ. This means that the level of detail a model should have to accurately reproduce the DRPs expected must be agreed in advance.

# 1 Introduction

Applying expert knowledge in hydrology, as in any other natural science, is crucial for linking observations and laws governing a given system, such as a catchment. It usually involves formulating and testing hypotheses about how the system functions (Savenije, 2009). At the root of this scientific reasoning, two opposing philosophies can be identified: the top-down and the bottom-up approaches. The first can be traced back to the Greek philosopher Plato (428 – 348 BC), who was trying to link general theories about the functioning of complex systems to measurable observations. A "bottom-up" approach involves extrapolating general theories from given observations, and can be attributed to Plato's student Aristotle (384 – 322 BC). These two approaches have been applied in nearly all scientific disciplines, e.g. in mathematics (Cellucci, 2013), economics (Böhringer and Rutherford, 2008) and neuroscience (Gilbert and Li, 2013), as well as hydrology. Thus one type of hydrological scientist, the experimentalist or "wet" hydrologist, tries to understand catchment functioning through extended field investigations, whereas the modeller or "dry" hydrologist tends to develop theories at the catchment scale and successively tries to validate them against measurements (Seibert and McDonnell, 2002).

Both modellers and experimentalists agree on the importance of expert knowledge for improving the realism of hydrological models, e.g. by forcing the model to reproduce the processes observed in the catchment. In recent years, several process-oriented approaches have been developed, of which the concept of dominant runoff process (DRP, for list of abbreviations see Table A1) is one (Blöschl, 2001). It relies on the hypothesis that, among the different runoff generation mechanisms that may occur at a given location (Hortonian overland flow HOF, saturation overland flow SOF, subsurface flow SSF, and deep percolation DP), one, the DRP, will be dominant over the others. Based on this concept, the following process-based modelling chain has been proposed (Clark et al., 2015): (i) reading the landscape, identifying and classifying the processes, (ii) developing a proper parameterisation to reflect our perceptions of the processes observed and (iii) allocating the parameter values of these parameterisations (Fig. 1).

Wet and dry hydrologists disagree, however, on how to implement their expert knowledge in each of these steps. For example, Schmocker-Fackel et al. (2007) applied the two philosophies to hydrological classifications using DRPs and claimed: "[…] These top-down approaches try to identify homogeneous landscape units. The assumption is that the hydrological response will also be homogeneous. By contrast, in bottom-up approaches, runoff formation is investigated on the plot scale and then units with the same runoff forming process are identified" (Schmocker-Fackel et al., 2007). Examples of such bottom-up mapping approaches can be found in Markart et al. (2004); Smoorenburg (2015), Scherrer AG (2006), Scherrer and Naef (2003) and Tilch et al. (2006), and of top-down mapping approaches in Gao et al. (2014), Gharari et al. (2011) and Fenicia et al. (2016).



Different interpretations of the two philosophies have been applied in hydrological modelling. For example, Hrachowitz and Clark (2017) maintain bottom-up models correspond to physically-based models, where the conservations laws on mass, momentum and energy are solved. In contrast, top-down models are conceptual models. With regard to the level of modelling detail, Nalbantis et al. (2011) linked monometric approaches, where some components are examined in detail and

other ones are only roughly described, to the bottom-up philosophy and the holistic approach, when all components are modelled with the same degree of detail, to the top-down one. Sivapalan et al. (2003), in contrast, classify approaches according to the scale considered: if the modelling is performed first at the small scale of e.g. HRU, or hillslopes, and then the results are scaled up to the catchment scale, it can be defined as bottom-up, whereas lumped models developed directly at the catchment scale can be defined as top-down. The definition of Sivalapan et al. (2003) also works well with the concepts

of model parameterisation and parameter allocation. For example, in a classical bottom-up exercise, parameter ranges are narrowed and/or model parameterisations are proposed based on catchments properties, expert knowledge and possibly inferences from measurements. By following a top-down approach, expert knowledge can be used instead to define relational rules between the parameters and fluxes of different landscape classes. In this way, the model is forced to behave according to the modeller's perception of the catchment functioning and the parameter space can be reduced so that no

calibration is necessary (Bahremand, 2016; Gharari et al., 2014).

Both approaches have strengths and weaknesses when implementing expert knowledge in process-based hydrological modelling. Bottom-up mapping approaches are often considered to require much data (Hümann and Müller, 2013; Müller et al., 2009), whereas top-down classification approaches are considered too coarse to detect the spatial distribution of processes with enough accuracy (Antonetti et al., 2016). Top-down models and parameterisations may be too simplistic and,

therefore, require calibration (e.g. Fatichi et al., 2016), whereas physically-based models may be too data demanding and not flexible enough to cope with emergent patterns at large scales (Beven, 2000).

Several attempts have been made to combine bottom-up and top-down philosophies (e.g. Klemeš, 1983; Sivapalan et al., 2003), and Hrachowitz and Clark (2017) in particular stress the need to merge forces. Similarly, Clark et al. (2017) ask: "How can we combine different perspectives on hydrologic modelling to advance the quest for physical realism?". Related

questions concern the level of detail needed to reproduce the observed dynamics and pattern and how much detail the available data warrants for a meaningful parameterisation of the chosen process representation (Clark et al., 2015). Clark et al. (2016) note that the structure of the model should reflect that of the landscape. They claim that focussing on the extent to which space accounting models are limited by the available data helps testing the mapping theories and, consequently, improves how well landscape details are represented in models.

Several frameworks have been proposed for testing working hypothesis (e.g. Best et al., 2011; Fenicia et al., 2011; Kraft et al., 2011), but few addressed these questions and explicitly consider ways of implementing expert knowledge in hydrological models. For example, McMillan et al. (2011) developed a set of diagnostic tests based on field data to formulate recommendations for model building. Contextually, Clark et al. (2011) used the modelling framework FUSE (Clark et al., 2008) to allow a proper model structure to be selected based on these recommendations. However, the use of flow data to



formulate the recommendations restricts the application of this method to ungauged basins (Hrachowitz et al., 2013). In addition, both the proposed recommendations and the FUSE framework are applicable exclusively at the lumped catchment scale. As a further development of FUSE, Clark et al. (2015) developed the SUMMA approach to provide a framework for both modellers and experimentalists to test alternative model discretisations, parameterisations, and numerical schemes.

Nalbantis et al. (2011) compared a bottom-up and a top-down modelling approach with a focus on catchments with high human impact.

Our study is intended to explore how different ways of implementing expert knowledge in hydrological modelling can affect simulation results with a specific focus on floods. In particular, we investigated: (i) Whether the use of more expert knowledge during the mapping phase improves hydrological simulations. (ii) Under which conditions (event type, catchment

characteristics) satisfying results can be reached without drawing much on expert knowledge during the mapping phase? (iii) How uncertainty in forcing data and in the initial conditions influences and/or interacts with the simulation results? (iv) How the model setup, i.e. the parameterisation approach and the parameter allocation strategy, affects the results?

To address these questions we produced so-called process maps of a mesoscale catchment in the Swiss Pre-Alps using three mapping approaches derived with different levels of involvement of experts. The effects of the differences between the

process maps on runoff simulations were investigated with two different setups of the newly developed PROcess-based Runoff Generation Module (RGM-PRO; Antonetti et al., 2017), which was forced with input data of varying quality. Finally, an analysis of variance (ANOVA) was performed to quantify the uncertainty arising from forcing data, process maps, model parameterisation and parameter allocation strategy.

## 2 Methods and Data

### 2.1 Study Area and Process Maps

We performed simulations on the Emme catchment up to Emmenmatt (445 km$^2$), which is located in the Pre-Alps mainly in Canton Bern and, on the eastern side, in Canton Lucern (Fig. 2). Its elevation ranges from 638 to 2213 m a.s.l.. About half of the catchment (52%) is covered by meadows, and the remaining part by forests (44%) or settlements (4%). The upper part of the catchment is characterised by Flysch and Cretaceous, whereas Freshwater and Marine Molasse and, to a lesser extent,

Moraine dominate the lower part of the basin. Three additional runoff gauging stations can be found in Eggiwil (Emme catchment, 125 km$^2$), Langnau i.E. (Ilfis catchment, 184 km$^2$) and Trubschachen (Trueb catchment, 55 km$^2$), and their measurements were used for this study to evaluate the performance of the models.

The study catchments were mapped according to three approaches with different levels of expert involvement and differing in terms of the data and the time required for mapping (Table 1 and Fig. 3). The simplest mapping approach includes solely

topographical information and distinguishes three landscape classes, i.e. wetland, hillslope and plateau, by combining the Height Above the Nearest Drainage (HAND) descriptor (Rennó et al., 2008) and slope (Gharari et al., 2011). These classes are supposed to be a proxy for saturation overland flow (SOF), subsurface flow (SSF), and deep percolation (DP). The expert





knowledge involved in this mapping approach is restricted to verifying the classification criteria. We refer to the process maps derived with the Gharari et al.'s (2011) approach as GH11 maps. Müller et al. (2009) developed classification criteria that take into account the topography (slope), land use and permeability of the geological substratum where again expert knowledge is only required for verification phase. This results in nine output classes, where, beside the DRP, information on

the process intensity is provided with a number from "1" (almost immediate reaction) to "3" (strongly delayed reaction). As the classification was developed by optimising the classification criteria against a reference map, the method can be also seen as top-down. The resulting process maps are referred to as MU09 maps.

Such simplistic, top-down mapping approaches have been criticised by experimentalists for finding no direct relationships between the runoff coefficient and slope (e.g. Scherrer, 1997). The third mapping approach we used is based on the

experimentalist approach introduced by Schmocker-Fackel et al. (2007) and Margreth (2010), which has already been used in, for instance, Antonetti et al. (2016) and Antonetti et al. (2017). Basically, the approach consists of the following steps. (1) All the available information about a given catchment, including its topography, land use, vegetation, soil, geology, and hydrogeology, is collected and the classification algorithm adapted accordingly. (2) Small test areas are identified and manually mapped according to Scherrer AG (2006). (3) The parameter values of the algorithm are identified by comparing

the automatically derived map with that derived manually on the test area. (4) Locations where estimations are not straightforward are verified with a field survey and, where necessary, adjustments are carried out. (5) Step (4) is reiterated until the process map is considered to be consistent with reality. Expert knowledge plays a crucial role in this bottom-up method, as all the experimentalists' detailed and qualitative knowledge about processes can be drawn on in the mapping. To reduce the number of resulting classes, the DRPs of MU09 maps and SF07map were reclassified into five different runoff

types (RTs) according to the intensity of the contribution to runoff (Table 2).

## 2.2 The Runoff Generation Module RGM-PRO

The implementation of a physically-based hydrological model was beyond the scope of this study even though the goal was to combine bottom-up and top-down approaches at each step in the modelling chain. This could be seen to go against the

definition of bottom-up model favoured by Hrachowitz and Clark (2017) and others, who associate it with physically-based. The concepts "bottom-up" and "top-down" can, however, be interpreted differently even if applied to the same topic and some researchers recommend using a semi-distributed conceptual model to accommodate the features of a catchment efficiently (Savenije and Hrachowitz, 2017). To perform the hydrological simulations for this study the newly developed conceptual PROcess-based Runoff Generation Module (RGM-PRO) was therefore used (Antonetti et al., 2017).

RGM-PRO has a grid based discretisation and was applied with a grid size of 500 m. It is able to take into account the sub-grid variability of the output classes of the process maps (Fig. 4). The model is structured so that a specific combination of storages can be defined for each output class of a given hydrological classification, with one storage system for the plant-available soil moisture (SSM), one for the runoff generation (SUZ) controlled by four parameters, and a third for



groundwater (SLZ; cf. Gurtz et al., 2003;Viviroli et al., 2009b). The separation of rainfall between the storage of plant-available soil moisture and the runoff generation module is controlled by a non-linearity parameter (BETA) fixed here at a value of 3 (Viviroli et al., 2009a). In SUZ, the storage times for overland flow (K0H) and subsurface flow (K1H) regulate the generation of the runoff. A threshold (SGRLUZ) determines the separation between overland and subsurface flow,

whereas a maximum percolation rate (CPERC) controls the percolation to the groundwater storage. This is divided into one quick-leaking and two slow-leaking storages controlled by three parameters (SLZ1MAX, CG1H, and K2H). For a more detailed description of the groundwater storage system, see Viviroli et al. (2009b) and Schwarze et al. (1999). This basic structure can then be adapted according to the features of the output classes of a given hydrological classification.

### 2.2.1 Model initialisation

The initial conditions can significantly affect simulation results (Liechti et al., 2013). For example, in a study about the uncertainties involved in operational flood forecasting chains in an alpine Swiss catchment, Zappa et al. (2011) found that uncertainty in initial conditions lasts for the first 48 hours, but is almost negligible compared with the uncertainty originating from meteorological data. To investigate to which extent the initial wetness conditions of a catchment affect simulation results with the event-based RGM-PRO, information on the plant-available soil moisture is assimilated from quasi-

operational grid-based simulations of PREVAH with a resolution of 500 m (Zappa et al., 2014). At the beginning of the simulations, therefore, the soil moisture value simulated with the PREVAH hydrological system was assigned to each output class of the corresponding cell. Alternatively, as the spatial variability of the soil moisture is higher than the model resolution (500 m), the hydrological downscaling technique described in Blöschl et al. (2009) and used in Antonetti et al. (2017) was implemented. The technique relies on three assumptions: (i) the soil moisture pattern at the smaller scale is time invariant,

which allows the process maps to be used as fingerprint; (ii) the spatial variance of the soil moisture at the smaller scale is linked with the one at the larger scale by a scaling theory taken from literature (Blöschl et al., 2009); and (iii) the soil moisture is mass conserving. After the soil moisture was downscaled to a resolution of 25 m, it was successively re-aggregated to obtain an average value for each output class and for each grid cell. Although no expert knowledge is directly involved in this step, the influence of the downscaling technique on the results was still investigated.

### 2.3 Parameterisation and Parameter Allocation Strategies

Our investigation focussed on floods, where the main processes to be parameterised are the runoff generation within the catchment, the runoff concentration to the drainage net and runoff routing in the stream channel. According to Sivapalan et al.'s (2003) definition, in a bottom-up modelling experiment these three steps are generally parameterised in an explicit manner in the model (Fig. 5). For example, runoff concentration can be taken into account by using a lag function, a linear

storage or a combination of them (e.g. Nash, 1957). In a similar way, runoff routing can be considered with a hydraulic approach (for a review, see Heatherman, 2008) or a simpler method such as linear storage in the so-called hydrological approach. Conversely, in a top-down configuration, runoff generation, concentration and routing do not necessarily have to



be treated separately (Fig. 5). In both the bottom-up and top-down parameterisations, a consistent parameter allocation strategy was implemented as described in the following sections.

### 2.3.1 Bottom-up setup: A priori definition of parameter ranges

For the bottom-up setup, RGM-PRO was configured as in Antonetti et al. (2017). The main catchment was first subdivided into several sub-catchments up to 2 km$^2$ in area. The runoff concentration to the outlet of each sub-catchment was therefore explicitly modelled for both overland and subsurface flow. For overland flow, the flow times were calculated using a semi-hydraulic approach (Schulla, 1997), and for subsurface flow a linear storage with one single parameter (GS1H, a storage time) was used. The flow times for the runoff routing in the channel were calculated with a Strickler coefficient of 30 m$^{1/3}$ s$^{-1}$ (Schulla, 1997).

For the allocation of parameters, plausible value ranges were defined a priori for each parameter of RGM-PRO based on the results of sprinkling experiments, on physical properties of soils, and on expert knowledge (Table 2, see Antonetti et al., 2017). By optimising these initial ranges against generalised response curves for each runoff type, they were then further narrowed before being applied to the catchments. As the response curves refer exclusively to the total runoff, the parameter ranges were defined in a manner that allows overland flow and subsurface flow to be partitioned in different ways, provided that the total contribution to runoff reflects that of the corresponding response curve. The number of output classes of the process map by Gharari et al. (2011) differs from that of the process maps used in Antonetti et al. (2017) for the identification of plausible parameter ranges. However, by comparing the landscape classes and runoff types on two catchments on the Swiss Plateau using similarity measures, Antonetti et al. (2016) found out that the most similar pairs were wetland-RT1, hillslope-RT3, and plateau-RT5. The same initial ranges of these runoff types were therefore assigned to the corresponding landscape class accordingly.

### 2.3.2 Top-down setup: Parameter and Process Constraints

The storage constants for overland flow (K0H) and subsurface flow (K1H) in a top-down approach are expected to represent all three steps of the runoff process described above, i.e. runoff generation, concentration and routing, as in the PREVAH hydrological model (Viviroli et al., 2009a). For the parameter allocation, the initial ranges were defined for each parameter and each output class of the hydrological classification according to Viviroli et al. (2009b), who identified a range of suitable values for each parameter of PREVAH for flood predictions in ungauged mesoscale Swiss catchments (Table 3).

In addition, the model parameter were forced to respect the following constraints:

$$\vartheta_{RT1} < \vartheta_{RT2} < \vartheta_{RT3} < \vartheta_{RT5} < \vartheta_{RT5}$$
$$\vartheta_{WETLAND} < \vartheta_{HILLSLOPE} < \vartheta_{PLATEAU} \tag{1}$$

Where $\vartheta$ represents each parameter of RGM-PRO, namely SGRLUZ, K0H, K1H, and CPERC. For those parameters of RGM-PRO physically similar to those of FLEX-Topo, the same constraints as those imposed by Gharari et al. (2014) were defined for the three landscape classes wetland, hillslope, and plateau. For example, the threshold for the activation of





overland flow SGRLUZ was forced to be lower for wetlands, which have a lower storage capacity than the two other landscape classes of the GH11 maps. Similarly, the storage times for both overland and subsurface flow were set to be higher for plateaus than for hillslopes, which were in turn set higher than those for wetlands. The only exception was the storage time for the subsurface flow K1H for wetland (SOF) and plateau (DP). This was set at 1000 h as no subsurface flow was expected there according to hydrologists' understanding of SOF and DP. Similarly, the maximum percolation rate CPERC was forced to be higher for plateaus than for hillslope and wetlands. As the overland flow is expected to be faster than subsurface flow independent of the landscape class, the constraint between the two storage times were defined as follows:

$$K0H_i < K1H_i \tag{2}$$

Where the subscript $i$ refers to the output classes of the GH11 maps, namely wetland, hillslope, and plateau. Following the same reasoning, parameter constraints were defined for the five runoff types of the SF07 and MU09 maps, i.e. RT1-5 (Eq. 1-2). One process constraint in addition to the parameter constraints was defined, namely that the specific peak runoff ($qmax$) should be higher for faster runoff types (Eq. 3):

$$qmax_{RT1} > qmax_{RT2} > qmax_{RT3} > qmax_{RT4} > qmax_{RT5} \tag{3}$$

or for landscape classes (Eq. 4):

$$qmax_{WETLAND} > qmax_{HILLSLOPE} > qmax_{PLATEAU} \tag{4}$$

Randomly selected parameter sets satisfying the parameter constraints were used to run the modelling chain combinations in the top-down setup. After the simulations, the runs also satisfying the process constraint were then used for the model evaluation, whereas the other runs were discarded (Gharari et al., 2014).

## 2.4 Experimental Design

To address the research questions, a total of 60 modelling chain combinations were designed (Fig. 6). To investigate the interaction between expert knowledge and quality of forcing data, meteorological data with increasing levels of accuracy were used. Precipitation data from five automatic stations in or close to the basin with a hourly resolution were interpolated based on Thiessen polygons (Thiessen, 1911) and following an Inverse Distance Weighting (IDW) method (Isaaks and Srivastava, 1989) with the power parameter p set equal to 2. In addition, the Combiprecip product (Sideris et al., 2014), a combination of ground measurements and radar quantitative estimations of precipitation, was used. To gradually increase the degree of realism, different spatial aggregations of Combiprecip were introduced. First, for each time step, the average precipitation intensity was distributed all over the main basin (CPC.mean). In the next configuration (CPC.mean.subc), the average precipitation intensity was calculated for and assigned to the corresponding sub-catchment. Finally, the Combiprecip data were used directly as they were delivered by MeteoSwiss. A total of six events were simulated with each modelling chain combination (Table 4). According to the flood type classification of Sikorska et al. (2015), three of them can be classified as short-duration events, and the remaining three as long-duration events. The event in August 2005 was also considered in this study even though no data from the automatic meteorological stations were available, as it was by far the largest flood event to have taken place in the last decades in Switzerland (Hegg et al., 2008).



At the beginning of each simulation, for each grid cell, the spatially distributed soil moisture data from PREVAH simulations were either directly assigned to each output class, i.e. runoff type or landscape class, or first downscaled (section 2.2.1) and successively re-aggregated to obtain an averaged value for each output class from the process map. The three mapping approaches of increasing complexity described in section 2.1 were used to map the spatial distribution of the DRP areas. Finally, the two parameterisations of section 2.3 were applied, each with its own parameter allocation strategy. For the modelling chain combinations based on the bottom-up setup, 10 different combinations of parameter values were randomly selected within the ranges defined a priori (section 2.3.1) to gain insights into the parameter uncertainty. For each modelling chain based on the top-down setup, a Monte Carlo simulation with 100 runs was performed for the same reason. To make comparison fairer, however, only the first ten combinations satisfying the process constraint were considered. For both setups, the value distribution within each range was assumed to be uniform.

The modelling chain combinations forced with the best quality and most realistic data, i.e. those driven with Combiprecip data and hydrological downscaled soil moisture data, were treated as the benchmark modelling chains.

Simulations were evaluated with the Kling Gupta Efficiency (KGE; Gupta et al., 2009):

$$KGE = 1 - \sqrt{(r-1)^2 + (\alpha-1)^2 + (\beta-1)^2} \tag{5}$$

This allows not only the correlation between the simulated and measured runoff ($r$) to be taken into account, but also the ratio between the standard deviation of the simulated runoff and that of the measured runoff ($\alpha$), and the ratio of the mean simulated to the mean observed discharge $\beta$. Furthermore, to quantify any potential overconfidence problems with the model setups, two factors were calculated, the P-factor and the R-factor (Abbaspour et al., 2009). The P-factor is the fraction of the measured runoff enveloped by the uncertainty band originating from the different runs of the Monte Carlo simulations, whereas the R-factor is the average width of the uncertainty band divided by the standard deviation of the measured runoff. Ideally, the P-factor is equal to 1, meaning that the observed hydrograph is bracketed by the model parameter uncertainty, whereas the R-factor tends to be zero, i.e. the simulation has the smallest uncertainty band.

Finally, to obtain insights into which uncertainty source contributes most to the total predictive uncertainty, an analysis of variance (ANOVA) was carried out. Compared to other sensitivity analysis methods, ANOVA was found to yield the most robust results without much computational efforts (Tang et al., 2007). ANOVA is based on the assumption that the uncertainty of an environmental system can be explained by the output variance generated by different effects, and has already been used to assess uncertainty, for instance, in climate impact projections (Addor et al., 2014; Bosshard et al., 2013; Köplin et al., 2013) and agro-hydrological applications (Moreau et al., 2013). ANOVA helps to clarify the question of how much of the available expert knowledge is worth feeding into a hydrological classification, given the unavoidable uncertainty linked with the input data. Assuming that all the chain components have an effect on the variability of the simulation performance $\Delta KGE$, the following effect model was used:

$$\Delta KGE = \overline{KGE} + ID_a + IC_b + PM_c + PP_d + I_{abcd} + \varepsilon_{abcd} \tag{6}$$



Where $\overline{KGE}$ represents the mean performance of the modelling chain combinations, $ID_a$ is the main effect of the input data ($a$ = THI, IDW, CPC.mean, CPC.mean.subc, CPC), $IC_b$ is the main effect related to the initial conditions ($b$ = with and without hydrological downscaling), $PM_c$ is related to the process maps with increasing amount of expert knowledge ($c$ = GH11, MU09, and SF07), and $PP_d$ to the parameterisation and parameter allocation approaches ($d$ = bottom-up, and top-

down). $I_{abcd}$ represents the interactions between the main factors and $\varepsilon_{abcd}$ the residual error. Each effect is checked for its representativeness and only those with a p-value lower than 0.05 are taken into account (Chambers et al., 1992).

## 3 Results

Using the benchmark modelling chain (i.e. Combiprecip and downscaled initial soil moisture data) and varying the process maps produced different results on the catchments investigated, depending on the model setup (i.e. parameterisation and

parameter allocation strategy) used. For example, in the Emme catchment up to Emmenmatt during the rainfall events of August 2005 (Fig. 7a) and September 2012 (Fig. 7b), the modelling chain based on the SF07 map simulated best the runoff peaks for the bottom-up setups, whereas the discharge volume was reproduced satisfactorily with all the process maps. However, irrespective of the process map used, the runoff peaks were simulated with a certain delay, and the falling limb of the hydrograph was overestimated, especially for the short-duration event. With the top-down setup, the modelling chain

based on the GH11 maps reproduced the runoff peaks better than the other process maps, whilst the runoff volume was slightly underestimated, independent of the process map used.

The results for the other simulated events in the catchments investigated were analysed to gain further insights into the effects of using process maps with different involvement of expert knowledge (Fig. 8). With regard to the short-duration events (Fig. 8a), the bottom-up outperformed the top-down setup in all the catchments investigated with the exception of the

Trueb sub-catchment, where none of the configurations reached satisfying results. Concerning the bottom-up configuration, SF07 maps performed best six times, i.e. slightly more often than the MU09 maps (four times), whereas GH11 never performed better than any of the other process maps. In contrast, when performed with the top-down parameterisation, the GH11 map obtained on average better results than the SF07 map, which, in turn, performed slightly better than MU09 map. With respect to the long-duration events (Fig 8b), the performance difference between the two parameterisations was

minimal on the main catchment (Emmenmatt), and on the Emme up to Eggiwil, whereas the combinations based on the bottom-up setup performed better than those based on the top-down setup on the Ilfis. None of the two parameterisations outperformed the other one on the Trueb sub-catchment, as they performed best once each. Similarly to what was observed for the short-duration events, none of the process maps outperformed the others within the bottom-up parameterisation. With regard to the top-down setup, the results obtained with the GH11 maps were on average better than those obtained with the

other process maps on Emmenmatt and Eggiwil, whereas the MU09 maps performed best on the Ilfis sub-catchment. Again, no clear trend emerged on the Trueb sub-catchment. Over all, the performance spread between different runs of the same Monte Carlo simulation was considerably higher for the top-down than for the bottom-up configuration. Among the



combinations based on the top-down experiment, the parameter uncertainty was found to be higher for GH11 maps than for the other process maps.

A visual inspection of the hydrographs in Fig. 9 shows that feeding the modelling chains with rainfall data spatially interpolated with Thiessen polygons has a considerable effect on the runoff peaks and, consequently, on the simulated runoff volume. However, no effect was detected for the falling limb of the hydrographs. Both model setups systematically underestimate the runoff at the gauging station of Trueb, independent of the process map used.

More generally, forcing the modelling chains with rainfall data of lower quality generally decreased the model performance (Fig. 10), moderately for the main catchment and more markedly for Eggiwil and for the Ilfis sub-catchments. The Trueb sub-catchment is an exception, as the use of rainfall data of lower quality increased the model performance nearly everywhere, independent of the process map used. Averaging the Combiprecip data over the whole catchment (CPC.mean) had the lowest impact on the simulated runoff, irrespective of the parameterisation approach and process map used. In contrast, using data interpolated with IDW and Thiessen polygons led on average to considerable performance losses, irrespective of the model parameterisation, especially for short-duration events. The performance losses for short-duration events were higher for the bottom-up than for the top-down setup, whereas their magnitude was similar among the two setups for long-duration events. The most pronounced performance changes were found in the Trueb sub-catchment with the bottom-up setup forced with Combiprecip data averaged over the sub-catchments. The choice of process map appeared to have little effect.

Uncertainty significantly increased with the decrease in size of the sub-catchments according to the analysis of variance (ANOVA), whereas the most important source of uncertainty was the parameterisation and parameter allocation strategy (Fig. 11a). The smallest source of uncertainty was the hydrological downscaling technique, which was found to be responsible for a slight improvement in simulation skills (Fig. A1). The influence of the process maps also increases with decreasing catchment size. However, when considering the two model configurations separately, the main uncertainty source varies depending on the catchment considered (Fig. 11b-c). With regard to the bottom-up experiment, the interaction between input data and process maps was found to be the largest source of uncertainty in the main catchment (Emmenmatt) and in the Ilfis sub-catchment. In the Eggiwil sub-catchment, the hydrological downscaling techniques and the input data were responsible for the largest uncertainties, whereas, on the Trueb sub-catchment, the process maps accounted for most of the differences in performance. Concerning the top-down setup, the input data were responsible for the largest variance in the main catchment and in the Eggiwil and Ilfis catchments, whereas the process maps were increasingly responsible for uncertainty with decreasing size of the sub-catchments.

## 4 Discussion

The main purpose of this study was to test different implementations of expert knowledge in a process-based hydrological modelling framework, following the basic assumption that combining top-down and bottom-up thinking can improve flood





predictions and potentially be applied in poorly gauged areas. Methods of different complexity were therefore tested for each step in the modelling process, including hydrological mapping, model parameterisation and parameter allocation. We wanted to find out whether the use of detailed expert knowledge during the mapping phase can improve simulation results, and how different levels of process knowledge interact with the model parameterisation and parameter allocation strategy when they

are forced by precipitation products of different quality. In the following sections, we discuss what light our findings shed on the research questions.

### 4.1 Can more expert knowledge in the mapping phase increase model performance?

We tested the hypothesis that a more complex mapping approach leads to better simulation results with a benchmark modelling chain forced with the best grid-based rainfall data available in real-time for the whole of Switzerland, that is the

10 Combiprecip product (Sideris et al., 2014). Recently, Antonetti et al. (2016) speculated on the added value of using as much of the available expert knowledge as possible for the hydrological classification. Our findings showed, the hypothesis can only be confirmed for the bottom-up setup, where the modelling chain combinations based on the most complex mapping approach (SF07) resulted in, on average, the highest performances in the study catchments. Conversely, no clear performance improvement was obtained by using SF07 maps with the top-down setup, irrespective of the event type

considered. The best performances obtained with by the top-down setup and the GH11 map are most likely attributable to the lower number of classes in the GH11 approach (three instead of five), which allowed the model to be more flexible and consequently the hydrographs to be better reproduced, but not necessarily for the right reason (Kirchner, 2006). In fact, the exclusive use of topographical information for the DRP mapping combined with the top-down setup has been shown to work only in the main catchment and in the sub-catchment of Eggiwil. This suggests that combining the mapping method of

Gharari et al. (2011) and the parameter allocation strategy of Gharari et al. (2014) is potentially worthwhile for specific types of catchment, especially those topography-controlled, whereas in other basins more complex mapping approaches need to be used (e.g. on Ilfis and Trueb). Fenicia et al. (2016) similarly found that a catchment classification based on geology led to better results than a classification based on HAND in the Attert catchment in Luxemburg.

The results obtained with the simplified mapping approaches (MU09 and GH11) were, on average, only slightly lower than

25 those obtained with the SF07 maps. Therefore, as the effort needed to derive the simplified maps is substantially lower, using one of the two top-down mapping approaches investigated here may be the best choice in terms of cost-benefit. However, this conclusion is not acceptable from an experimentalist point of view. The results may seem acceptable at the gauging stations, but the local representation of the DRP mapped would most likely differ from that expected by an experimentalist. Topography alone cannot furnish information about the storage and infiltration capacity of soils, as Scherrer et al. (2007)

pointed out. Therefore, the two top-down mapping approaches tend to overestimate the runoff contribution of steep slope and underestimate it on flat areas (Antonetti et al., 2016).

Modellers and experimentalists need to agree on what they mean by realism, and how much detail hydrologists should provide to achieve it. An exact reproduction of processes at the plot scale (e.g. exact localisation of macropores etc.) is of




course unfeasible due to lack of data, and even knowledge, and the high computational effort such a level of detail would require (Beven, 2001, 2000; Semenova and Beven, 2015; Weiler and McDonnell, 2004). No experimentalist would therefore expect this level of detail from a process-based model at the catchment scale. However, in our opinion, the hydrological community should aspire to develop models able to reproduce processes in a "realistic" way (i.e. in agreement with the

experimentalists' expectation), at least at the sub-catchment or, even better, at the hillslope scale. This should be a feasible goal, especially considering how new measurements techniques continue to be developed and existing ones refined (Savenije and Hrachowitz, 2017). Such high requirements will probably challenge the validity of simplified mapping approaches and highlight the added value of the more complex ones. The availability of measured data for smaller sub-catchments, where the results of the mapping approaches differed greatly (e.g. in the upper part of the Eggiwil sub-basin), could have better

emphasised the potential added value given by more accurate process maps. Future research will address this topic.

## 4.2 Bottom-up versus top-down model setup

Which model setup was more efficient in modelling the catchment systems investigated in this study? To answer this question, the model parameterisations and the parameter allocation strategies used are addressed separately.

The low performances of the top-down setups in simulating the short-duration events probably depend on the

parameterisation approach chosen. The coupled parameterisation of runoff generation, concentration, and routing could well be responsible for the insufficiently fast reaction to high precipitation intensity, as, for instance, fast subsurface flow is basically not allowed to occur. With the bottom-up parameterisation, the underestimation of the falling limb of the hydrograph highlighted by the visual inspection of the hydrographs of Fig. 9 is ascribable to the poor representation of the runoff concentration by the bottom-up setup. However, the adaptation of the model structure, e.g. by introducing a function

for the explicit consideration of the time lag due to the processes of runoff concentration and routing, was beyond the scope of this study.

Concerning the parameter allocation strategies, the very same low performances reached by the top-down setup during short-duration events could be also related to the modellers' tendency to set relational rules among parameter and fluxes of different classes. Although the definition of parameter and process constraints force the model to behave according to the

modeller's perception of the catchment functioning, the parameter space defined by the initial parameter ranges of Viviroli et al. (2009b) was apparently still too large to ensure high performances with only 100 Monte Carlo runs. On the other hand, the bottom-up parameter allocation strategy led to overconfidence problems, as the measured runoff was only partially enveloped by the uncertainty bands defined by the different runs of the Monte Carlo simulation (Fig. A2). This is directly ascribable to the definition of very narrow initial ranges for each parameter (Antonetti et al., 2017).

Considering the KGE deviations arising from the use of different forcing data furnished further insights into the setups tested here. The lower KGE deviations observed for the top-down setup showed that it can cope better than the bottom-up setup with uncertainties in the input data as it allows parameter values that can compensate for biases in the input data to be




selected. This also explains the larger performance spreads reached by the modelling chains based on the top-down setup, as not all the parameter sets fulfil the requirements for compensating a biased forcing.

The bottom-up setup is therefore suitable for identifying of uncertainty sources. Once the extent and distribution of DRPs on a given catchment corresponds to the experimentalist's perception, which may still be biased, and once, for each output class

of a process map, a proper parameterisation has been chosen, any remaining deviations of the simulated hydrograph from the measured hydrograph can be explained as arising from uncertainties either in the forcing data or in the measured discharge data.

### 4.3 Expert knowledge under uncertainty

The assumption that more reliable input data would have led to expert knowledge being more effectively applied in
hydrological classification was investigated by varying the forcing data of the different modelling chain combinations. No clear trend was however identified among the different process maps. Even using the CombiPrecip data used for the benchmark modelling chain, which provide the best spatially distributed estimation of rainfall data available in real-time for the whole Switzerland, led to considerable uncertainties, especially with short-duration events, due to its spatially resolution (1 km$^2$) and problems linked with radar images (see also Antonetti et al., 2017). When the input data are of low quality (e.g.
interpolated with simple approaches like IDW and Thiessen polygons), the way model performance can change is symptomatic of the presence of compensation effects within the model. For example, the largest deviations, which occurred in the Trueb sub-catchment, are attributable to the meteorological station on Napf, which is located at 1404 m a.s.l.. It only makes sense to regionalise the values from mountain stations if an elevation factor is taken into accounting, otherwise it may, as here, lead to a local overestimation of the precipitation and, consequently, of the discharge (Sevruk and Mieglitz,
2002; Sevruk, 1997).

Over the years, instead of refining the process maps by drawing on more knowledge in the mapping phase, the opposite occurred, and the uncertainty in the input data was used as an excuse for removing complexity from hydrological classifications. For example, Müller et al. (2009) developed their mapping approach based exclusively on information about topography, geology, and land use in order to simplify the method of Schmocker-Fackel et al. (2007), which is in turn a
simplification of the manual mapping approach developed by Scherrer and Naef (2003) and is based on all the information available about a basin. Only two years later, Gharari et al. (2011) introduced a further classification approach based exclusively on topography. This oversimplification risk could be avoided by defining better the minimal criteria for "realism" a model should fulfil before claiming that it had improved realism.

### 4.4 Quantifying uncertainty sources

The analysis of variance (ANOVA) on the catchments investigated showed that the uncertainty linked with parameterisation and parameter allocation strategies was always at least comparable quantitatively with that originating from the input data. For the sub-catchments investigated, it was even greater. This suggests that the step in the modelling process in question has





the highest potential for improvement. For two of the four catchments investigated, the uncertainty originating from the process maps was found to be comparable with that arising from the different input data. This means that, up to a certain catchment size, a proper mapping of processes is as important as the availability of reliable input data. The soil moisture data assimilated from PREVAH simulations could also represent an important source of uncertainty. Performing a virtual

experiment where the catchments were assumed to be completely saturated at the beginning of each event led to large overestimations of the initial peaks during an event (Fig. A3). However, with a view to an operational application of RGM-PRO, the data from the PREVAH simulations used in this study represent the best grid-based estimation of soil moisture available in real time (Horat, 2017). Using of soil moisture data from other grid-based models was beyond the scope of this study.

Results from the ANOVA also showed a considerable increase in uncertainty with decreasing size of the sub-catchments, as Hellebrand et al. (2011) also found and attributed to a wrong choice of the calibration catchment. The poor performances of the bottom-up setup in the Trueb sub-catchment, which originated the large uncertainty shown in Fig. 11, can, however, be attributed to the low quality of the measured discharge data. The measurement accuracy of the gauging station there has already been questioned in another study (Scherrer AG, 2012), and may of course compromise the potential benefits of using

more complex process maps. Checking the rating curve of the gauging stations was, however, beyond the scope of this study.

### 4.5 Limitations of this study

Some aspects to be investigated during future research include working towards a more thorough modelling system by investigating not only the runoff formation process but also other fluxes that can dominate in a basin such as

evapotranspiration and interception. Investigating the influence of expert knowledge on the parameterisation of these processes was beyond the scope of this study, but could represent a direction for future research. We restricted our modelling to an event-based runoff generation module because the SF07 maps and the MU09 maps had been developed with a focus on floods. The simulation time step of one hour for investigations on floods is limiting especially when simulating short-duration events (Steinbrich et al., 2016). Sideris et al. (2014) proposed a disaggregation scheme for the generation of

precipitation estimates with a resolution of five and ten minutes, but this involves still large uncertainties, and the hourly aggregated data was found to produce higher skill scores in the validation phase. We therefore only included hourly forcing in this study. The equations governing the storage behaviour were solved with an explicit Euler scheme, which has already been found to be responsible for uncertainty in other studies due to the numerical approximations involved (Kavetski and Clark, 2010). To address this issue, an adaptive number of sub-hourly integration steps was introduced according to the

intensity of water reaching the upper-zone runoff storage SUZ.

No soft data from experimentalists' campaigns was used to inform or validate our model. This approach was demonstrated to be valuable to pursue the dialogue between modellers and experimentalists (Seibert and McDonnel, 2002). For the evaluation of the modelling chain combinations, we used the KGE metric exclusively instead of multiple validation criteria



suggested by several authors (e.g. Güntner et al., 1999; Krause et al., 2005; Moussa and Chahinian, 2009; Seibert and McDonnell, 2002; Uhlenbrook and Leibundgut, 2002; Weiler and McDonnell, 2007). The KGE is, however, a comprehensive objective function that takes into account both peak and volumetric errors. It was therefore considered suitable for event-based model evaluation. Finally, to generalise the findings of this study, the number of catchments and

events investigated should be increased considerably. For example, investigating catchments with contrasting reactions to heavy rainfall should provide more support for using more complex mapping approaches to identify the extent and distribution of DRPs.

## 5 Conclusions

Recent calls to combine bottom-up and top-down reasoning to improve the realism of process-based hydrological models

were what motivated this study. We wanted to obtain insights into how to best use expert knowledge, given unavoidable uncertainties. First, we investigated how applying different degrees of expert knowledge in landscape classification affects the final outcome of hydrological simulations. We compared two different setups (i.e. parameterisation and parameter allocation strategies): the first is based on experimentalists' (bottom-up) reasoning, and the second is driven by a modellers' (top-down) thinking. We then looked at how performance varied with different levels of uncertainty in the forcing data

before finally quantifying the fraction of variance explained by each uncertainty source.

The main findings of the study were:

- Using complex process maps with high involvement of expert knowledge adds little potential value due to large uncertainties occurring even with the best forcing data available in real-time and in the measured discharge data. Performance using simplified mapping approaches was also satisfactory, especially for long-duration events.

- The bottom-up setup performed better on average than the top-down setup in the catchments investigated, independent of the process map used. The top-down setup was able to accommodate biases in the precipitation data at the expense of exactly identifying sources of uncertainty. Conversely, the bottom-up setup can be used diagnostically to identify uncertainty sources, but had overconfidence problems due to an overly narrow a priori definition of parameter ranges.

- The uncertainty linked with the process maps and, consequently, the importance of a realistic representation of the spatial distribution of processes, increased with decreasing size of the catchments.

In conclusion, modellers and experimentalists need to reach agreement on what they mean by "model realism", especially concerning the level of detail. In our opinion, a catchment scale model should be able to reflect the real distribution of dominant runoff processes up to the hillslope scale. More accurate process maps can help to achieve this goal.



## Data Availability

MU09 maps, GH11 maps and RGM-PRO are available from Manuel Antonetti (manuel.antonetti@wsl.ch), and the PREVAH soil moisture estimations from Massimiliano Zappa (massimiliano.zappa@wsl.ch). The GIS data used for deriving the MU09 maps and GH11 maps can be obtained undo license from the Federal Office of Topography swisstopo, whereas the SF07 maps were provided by Scherrer AG and SoilCom GmbH (contact the authors for help in accessing them). The runoff data is available from the Swiss Federal Office for the Environment and the Canton of Bern, and the precipitation data from the Swiss Federal Office of Meteorology and Climatology MeteoSwiss (free of charge for scientific purposes).

## Appendices

See Table A1, Fig. A1, Fig. A2, and Fig. A3.

## Competing interests

The authors declare that they have no conflict of interest.

## Acknowledgements

We thank the Swiss Federal Office for the Environment (FOEN) for funding the PhD project of the first author, the Swiss Federal Office of Meteorology and Climatology MeteoSwiss for providing the precipitation data, and FOEN and the Canton of Bern for providing the observed runoff series. We are also very grateful to Michael Margreth and Simon Scherrer for providing the SF07 maps of the catchments investigated and to Silvia Dingwall for language feedback.

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





**Table 1. List of the hydrological classifications used in this study, the data they require, the number of output classes used, and, in brackets, the number of output classes with the original approach. Adapted from Antonetti et al. (2016).**

| Abbr. | Authors | Topography | Land use | Geology | Soil maps | Extensive field investigations | Number of output classes |
|-------|---------|------------|----------|---------|-----------|-------------------------------|--------------------------|
| GH11 | Gharari et al. (2011) | X | | | | | 3 |
| MU09 | Müller et al. (2009) | X | X | X | | | 5(9) |
| SF07 | Schmocker-Fackel et al. (2007) | X | X | X | X | X | 5(12) |



**Table 2. Reclassification of DRPs in runoff types according to their contribution to runoff (HOF = Hortonian Overland Flow; SOF = Saturation Overland Flow; SSF = Subsurface Flow; DP = Deep percolation). 1 represents an almost immediate reaction, 2 a slightly delayed one and 3 a greatly delayed one. Adapted from Naef *et al.* (2000).**

| Runoff type | DRP | Runoff intensity |
|---|---|---|
| RT 1 | HOF1/2, SOF1 | Fast |
| RT 2 | SOF2, SSF1 | Slightly delayed |
| RT 3 | SSF2 | Delayed |
| RT 4 | SOF3, SSF3 | Greatly delayed |
| RT 5 | DP | Not contributing |



**Table 3. Parameter ranges for the top-down and bottom-up model configurations.**

| Bottom-up | | Runoff type | | | | | Landscape class | | |
|---|---|---|---|---|---|---|---|---|---|
| | | RT1 | RT 2 | RT 3 | RT 4 | RT 5 | Wetland | Hillslope | Plateau |
| SGRLUZ | [mm] | 0-40 | 40-100 | 40-100 | 100-200 | 200-400 | 0-40 | 40-100 | 200-400 |
| K0H | [h] | 0.05-0.4 | 0.05-0.4 | 0.05-0.4 | 0.05-0.4 | 0.05-0.4 | 0.05-0.4 | 0.05-0.4 | 0.05-0.4 |
| K1H | [h] | 1000 | 0.5-2 | 2-4 | 2-4 | 1000 | 1000 | 2-4 | 1000 |
| CPERC | [mm h$^{-1}$] | 0.1 | 0.1 | 0.1-0.5 | 0.5-5 | 5-50 | 0.1 | 0.1-0.5 | 5-50 |
| GS1H | [h] | | | | | 1-3 | | | |

| Top-down | | Runoff type | | | | | Landscape class | | |
|---|---|---|---|---|---|---|---|---|---|
| | | RT 1 | RT 2 | RT 3 | RT 4 | RT 5 | Wetland | Hillslope | Plateau |
| SGRLUZ | [mm] | 0-10 | 5-20 | 15-50 | 20-100 | 80-200 | 0-30 | 20-40 | 30-50 |
| K0H | [h] | 1-30 | 1-30 | 1-30 | 1-30 | 1-30 | 1-30 | 1-30 | 1-30 |
| K1H | [h] | 10-60 | 10-60 | 10-60 | 10-60 | 10-60 | 10-60 | 10-60 | 10-60 |
| CPERC | [mm h$^{-1}$] | 0.04-0.2 | 0.04-0.2 | 0.04-0.2 | 0.04-0.2 | 0.04-0.2 | 0.04-0.2 | 0.04-0.2 | 0.04-0.2 |





**Table 4. Start and end of the simulated events. IDW = Inverse Distance Weighting, THY = Thiessen Polygons.**

| Abbreviation | Simulation start | Simulation end | Event type according to Sikorska et al. (2015) | Specific peak runoff measured at Emmenmatt [$m^3\ s^{-1}\ km^{-2}$] | No. of stations available for IDW and THI |
|---|---|---|---|---|---|
| Aug10 | 29.07.2010 | 31.07.2010 | Short-duration | 0.48 | 2 |
| Sep12 | 11.09.2012 | 13.09.2012 | Short-duration | 0.40 | 5 |
| Aug14 | 11.08.2014 | 12.08.2014 | Short-duration | 0.61 | 5 |
| Aug05 | 19.08.2005 | 24.08.2005 | Long-duration | 1.08 | - |
| Jun12 | 07.06.2012 | 15.06.2012 | Long-duration | 0.19 | 5 |
| May16 | 11.05.2016 | 15.05.2016 | Long-duration | 0.34 | 5 |





**Table A1. List of abbreviations used in this study.**

| Abbreviation | Long name/description |
| --- | --- |
| ANOVA | ANalysis Of VAriance |
| BETA | Non-linearity parameter for infiltration module |
| CG1H | Storage time for quick baseflow |
| CPC | Combiprecip |
| CPC.mean | Combiprecip precipitation data averaged over the whole catchment |
| CPC.mean.subc | Combiprecip precipitation data averaged over the whole corresponding sub-catchment |
| CPERC | Maximum percolation rate |
| DP | Deep Percolation |
| DRP | Dominant Runoff Process |
| GH11 | Mapping approach after Gharari et al. (2011) |
| GS1H | Storage time for concentration of subsurface flow |
| HAND | Height Above the Nearest Drainage |
| HD | Hydrological Downscaling |
| HOF | Hortonian Overland Flow |
| IDW | Precipitation interpolated with the Inverse Distance Weighting method |
| K0H | Storage time for overland flow |
| K1H | Storage time for subsurface flow |
| K2H | Storage time for slow baseflow |
| KGE | Klingt-Gupta Efficiency |
| MU09 | Mapping approach after Müller et al. (2009) |
| P | Precipitation |
| PREVAH | PREecipitation-Runoff-EVApotranspiration HRU Model |
| Q | Discharge |
| R | Routing |
| RC | Runoff Concentration |
| RG | Runoff Generation |
| RGM-PRO | PROcess-based Runoff Generation Module |
| RT | Runoff Type |
| SF07 | Mapping approach after Margreth et al. (2010) and Schmocker-Fackel et al. (2007) |
| SGRLUZ | Thresold for overland flow |
| SLZ | Lower zone runoff storage |
| SLZ1MAX | Maximal content of the quick baseflow storage |
| SOF | Saturation Overland Flow |
| SSF | SubSurface Flow |
| SSM | Soil moisture storage |
| SUZ | Upper zone runoff storage |
| THI | Precipitation interpolated with Thiessen polygons |





# Modellers' (top-down) approaches

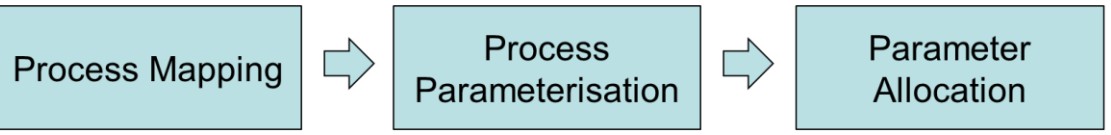

**Simplified mapping approaches**
- Müller et al. (2009)
- Gharari et al. (2011)

**Conceptual models**
- HBV (Bergström, 1976)
- PREVAH (Viviroli et al., 2009)

**Parameter and process constraints**
- Gharari et al. (2014)
- Hrachowitz et al. (2014)

Process Mapping ⇒ Process Parameterisation ⇒ Parameter Allocation

**Complex mapping approaches**
- Scherrer and Naef (2003)
- Schmocker-Fackel et al. (2007)

**Physically based models**
- CATFLOW (Zehe et al., 2001)
- WaSiM (Schulla, 1997)
- RoGeR (Steinbrich et al., 2016)

**Narrowing parameter ranges based on expert knowledge**
- Antonetti et al. (2017)

**Formulating recommendations**
- McMillan et al. (2011)

**Soft Data**
- Seibert and McDonnell (2002)

# Experimentalists' (bottom-up) approaches

**Figure 1: The three main steps for process-based flood predictions and the differences between the bottom-up (bottom) and top-down (top) approaches.**

**Figure 2: Maps of the Emme catchment, Switzerland. (a) Digital terrain model (25m resolution), river network and location of the runoff gouging stations; (b) land-use map (100 m resolution); (c) geology map. Data: BFS GEOSTAT/Federal Office of Topography swisstopo (DV033492.2).**





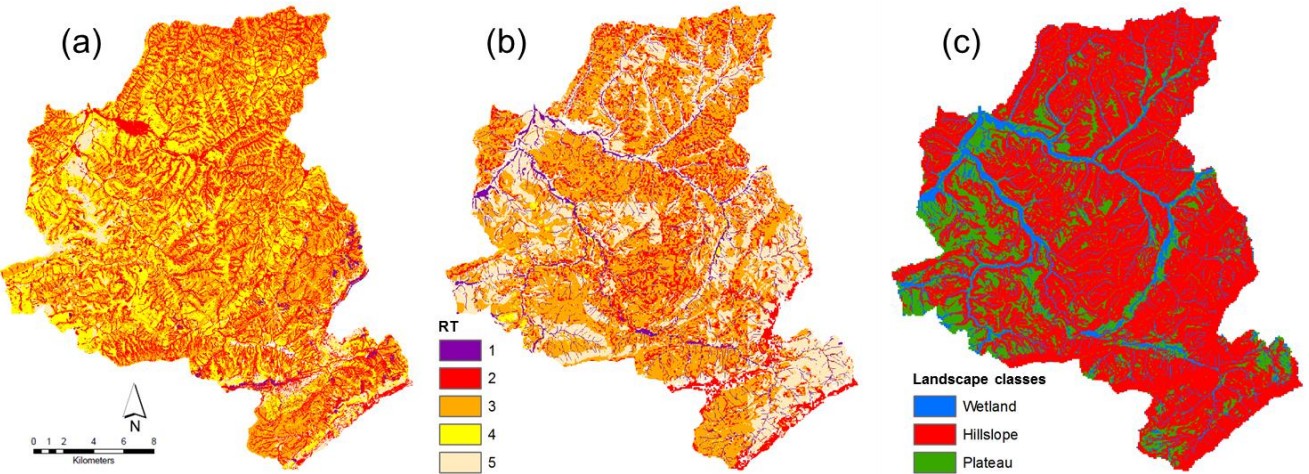

Figure 3: Process maps for the Emme catchment map according to (a) Schmocker-Fackel et al. (2007), (b) Müller et al. (2009), and (c) Gharari et al. (2011). RT = runoff type.





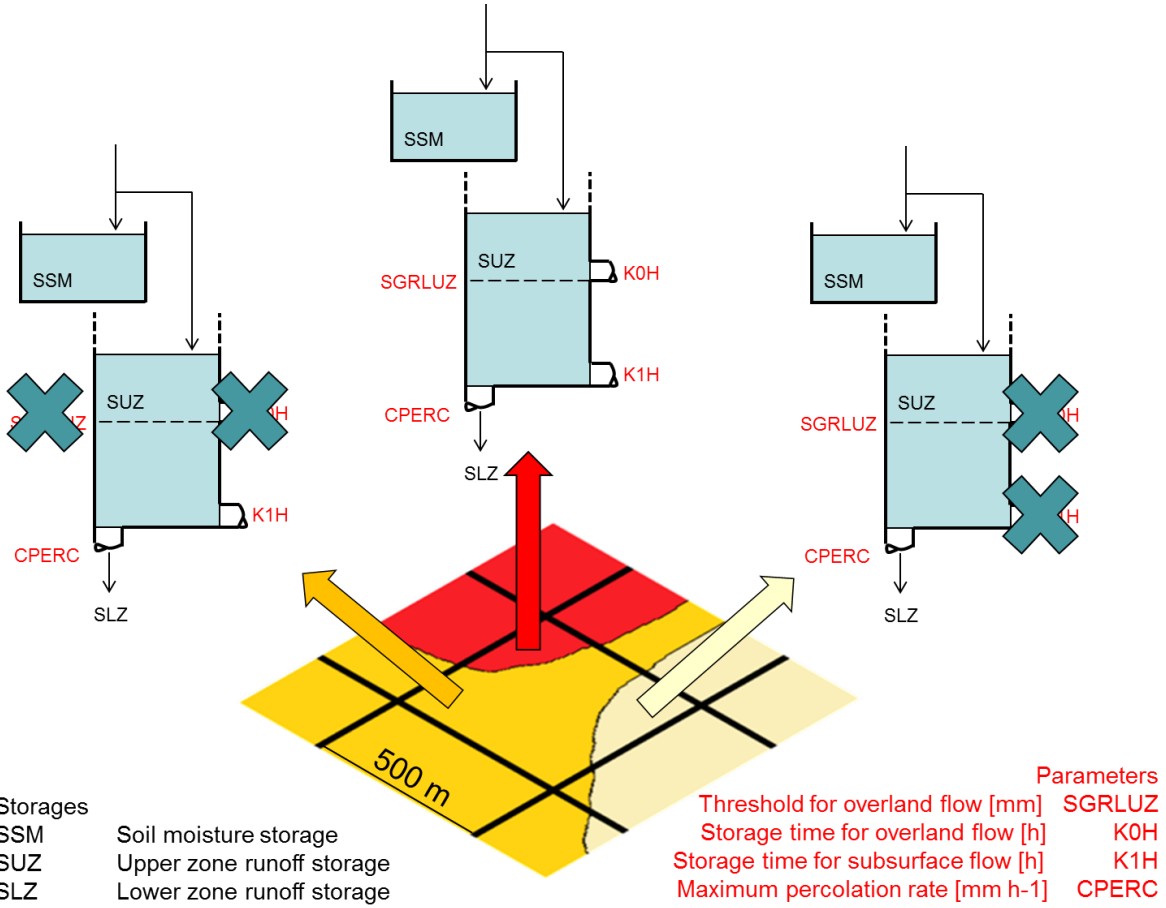

**Figure 4. Schematic representation of the spatial discretisation and structure of RGM-PRO. For each class of a given process map, a specific storage system can be defined. For list of abbreviations see Table A1.**





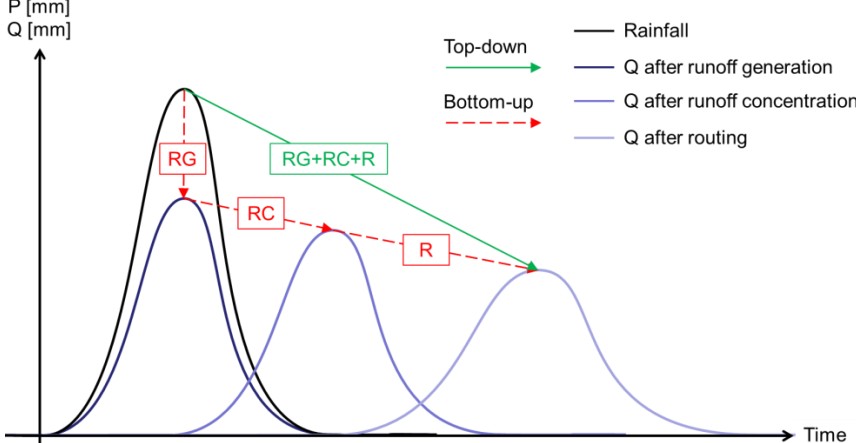

**Figure 5. Representation of runoff generation (RG), runoff concentration (RC) and routing (R) in the bottom-up (red) and in the top-down (green) setups. Adapted from Krebs et al. (2000).**





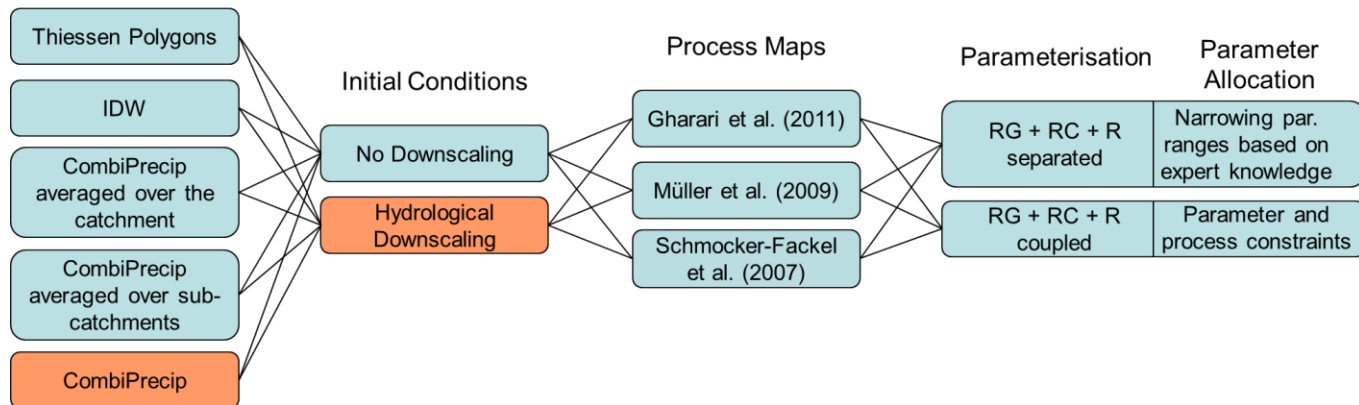

**Figure 6. Diagram of the modelling chain combination performed for this study. The components with an orange background form the benchmark modelling chain. IDW = Inverse Distance Weighting; RG = runoff generation; RC = runoff concentration; R = routing.**





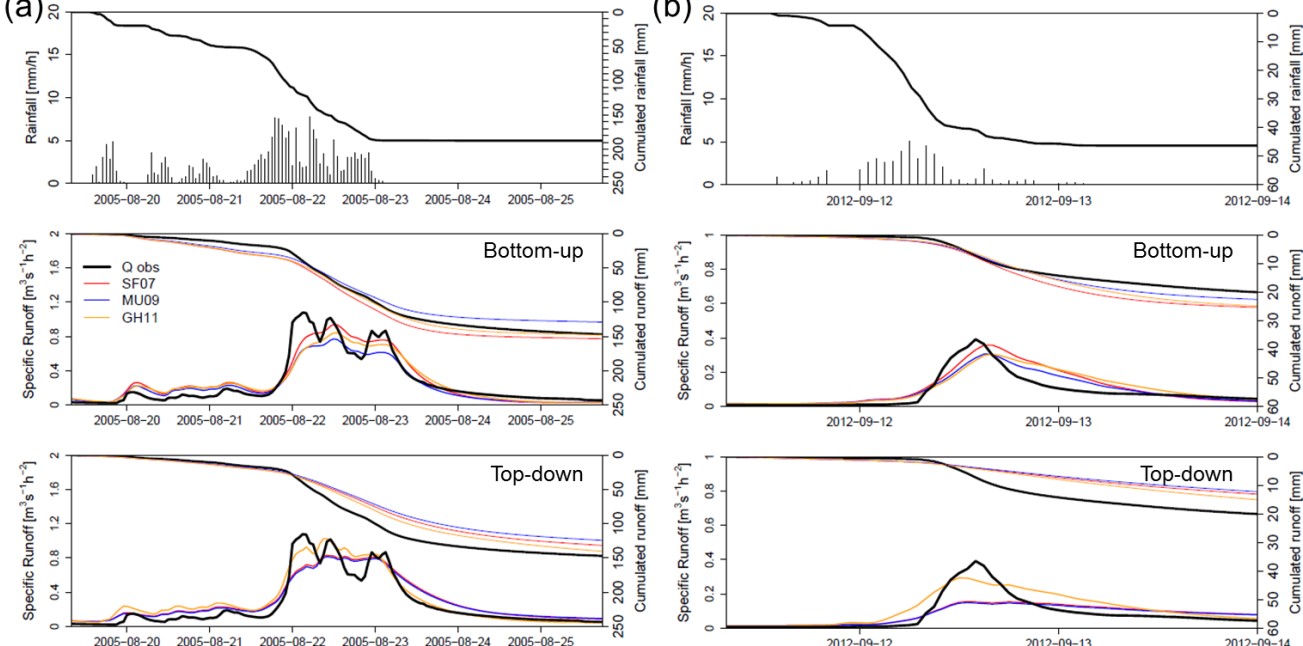

**Figure 7. Simulated runoff for the Emme catchment up to Emmenmatt during the long-duration event of August 2005 (a), and during the short-duration event of September 2012 (b), obtained from the different process maps and model parameterisations. The simulated hydrographs refer to the first run of the Monte Carlo simulation performed with the corresponding modelling chain combination. The SF07 map performed best with the bottom-up setup, whereas the GH11 map outperformed the other maps with the top-down setup.**



**Figure 8. Results from the short-duration (a) and from the long-duration events simulated on the catchments investigated using the benchmark modelling chain. The boxplots represent the simulation results of the bottom-up (white background) and of the top-down (grey background) parameterisations, whereas the coloured borders represent the different mapping approaches. Overall, the bottom-up performed better than the top-down setup during short-duration events, whereas no preference was found for long-duration events.**





**Figure 9. Simulated runoff for the four study catchment during the long-duration event of May 2016, obtained from different input data (CPC = Combiprecip; THI = Thiessen polygons), process maps (SF07, MU09 and GH11), and model setups (bottom-up and top-down). The simulated hydrographs refer to the first run of the Monte Carlo simulation performed with the corresponding modelling chain combination. Errors linked with the input data (e.g. the overestimation of the second runoff peak at Emmenmatt and Eggiwil due to a higher input signal) can be distinguished from those more clearly linked with the model parameterisation.**







**Figure 10. Averaged KGE deviations from the benchmark modelling chain (i.e. driven by Combiprecip data) obtained with the bottom-up (a) and top-down (b) configurations. Each block corresponds to a specific modelling chain based on the rainfall data reported on the left (CPC = Combiprecip; IDW = inverse distance weighting; THI = Thiessen polygons), whereas the displayed event types are reported at the top. The bars represent the average performance difference obtained from Monte Carlo runs for each of the four study areas, whilst the colour of the bars represent the different mapping approaches. Overall, the performance deviations were higher for the bottom-up than for the top-down setup.**





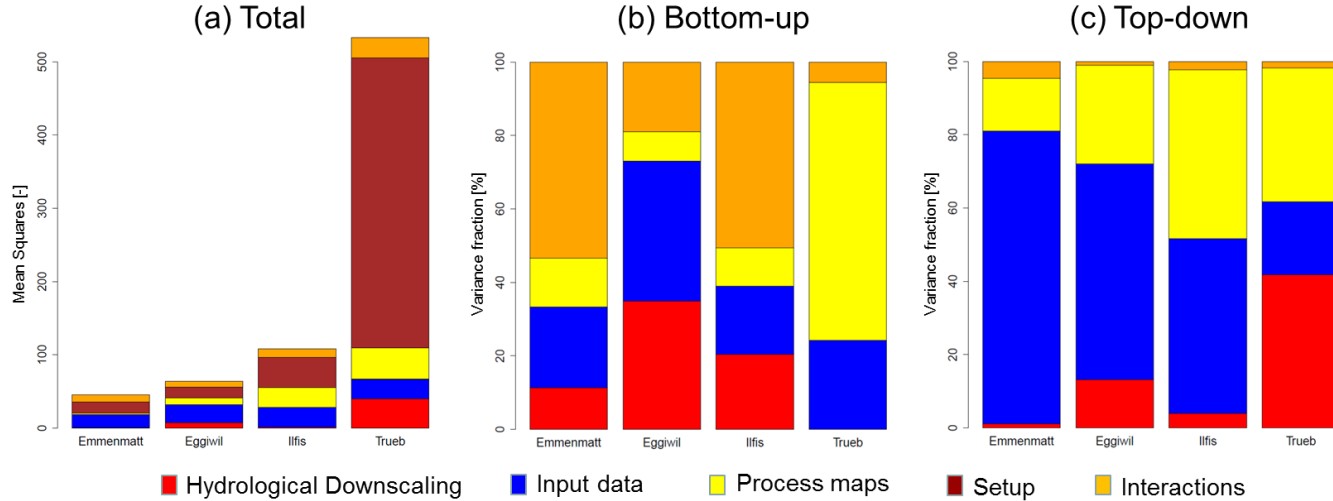

**Figure 11.** Decomposition of the model performance (KGE) variance at the four gauging stations for all the modelling chain combinations (a), as well as for those based on the bottom-up (b) and top-down (c) configurations. Total uncertainty increases with ingin size of the catchments.





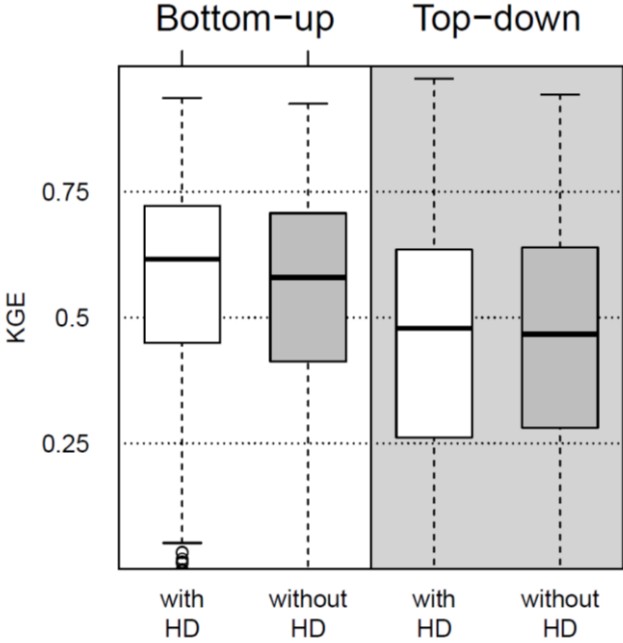

**Figure A1. Comparison of the simulation results obtained with and without hydrological downscaling (HD) of the initial conditions by the modelling chains based on either the bottom-up or the top-down configuration. HD slightly increased both the best and average performance of the model setups.**





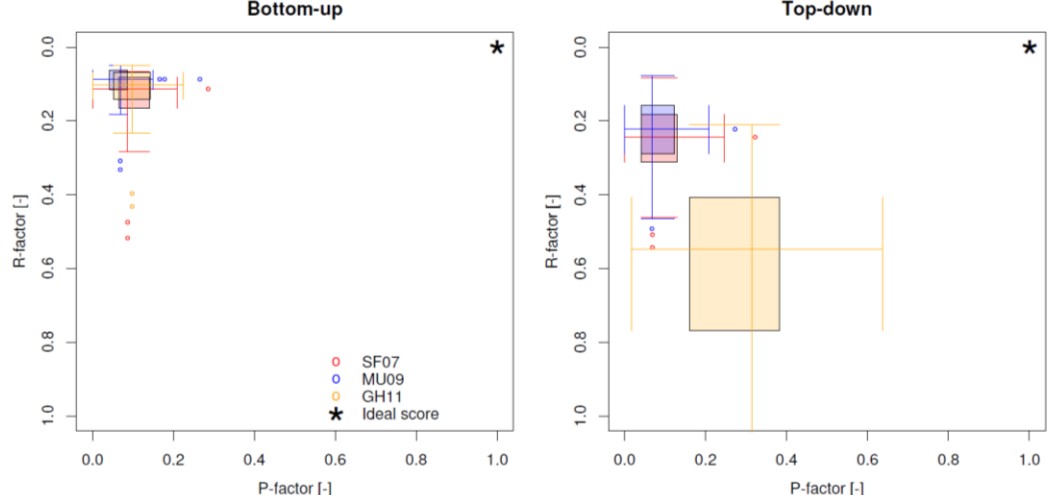

**Figure A2. Values of P-factors (x axis) and R-factors (y axis) calculated for the different process maps with the bottom-up and top-down setups. The ideal score (i.e. P-factor = 1 and R-factor = 0) is represented with a black asterisk. Whilst the process maps performed similarly with the bottom-up setup, the observed runoff was best bracketed by simulations obtained with the GH11 maps and the top-down setup, but at the expense of a wider uncertainty band (i.e. lower R-factors).**





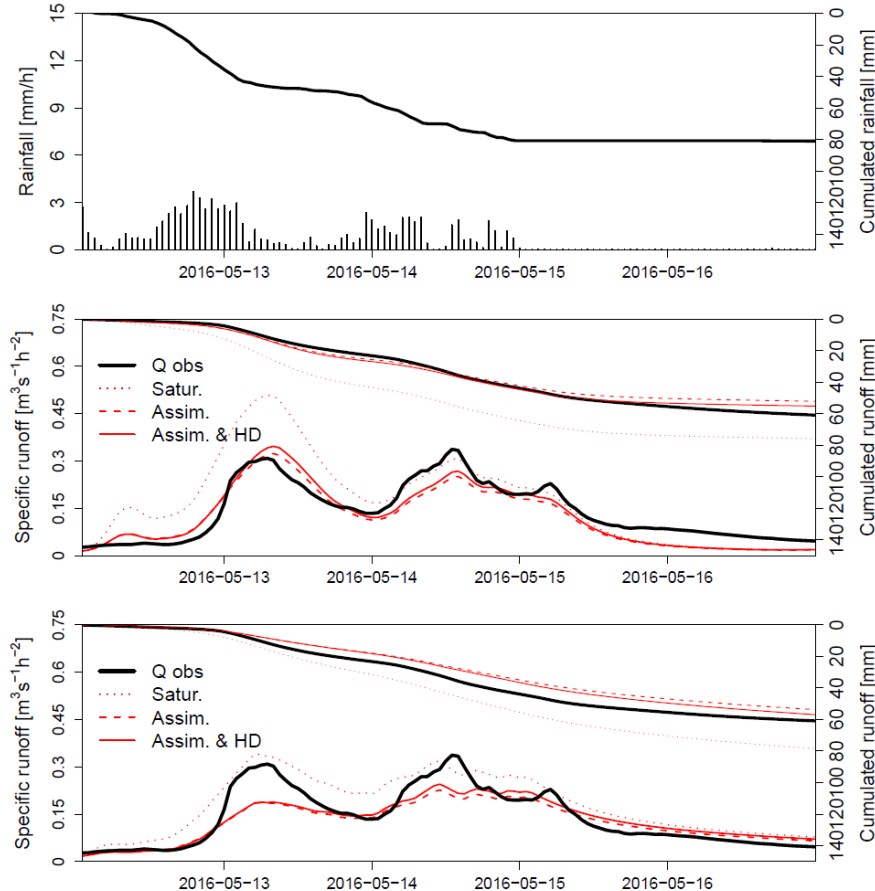

**Figure A3.** Influence of the soil moisture initial conditions on the simulated runoff for the Emme up to Emmenmatt during the long-duration event of May 2016 obtained with the bottom-up (a) and top-down (b) setup. The simulated hydrographs refer to the first run of the Monte Carlo simulation performed with SF07 map and with Combiprecip as forcing. The saturated initial conditions led to a significant overestimation of runoff at the beginning of the simulations, whereas the hydrological downscaling barely affected the simulation results.