# Peer review of "How can expert knowledge increase the realism of conceptual hydrological models? A case study based on the concept of dominant runoff process in the Swiss Pre-Alps"

_Hydrology and Earth System Sciences, 2017_

## Referee Comment (RC1) · Anonymous Referee #1 · 5 Jul 2017

The manuscript discusses the role of expert knowledge in hydrological modelling, contrasts several perspectives described as "modeller" and "experimentalist", and provides a simple case study illustrating the discussion.

I generally enjoyed reading the manuscript, it is written in an easy-to-follow style, and does not oversell its results. The case study is presented in a fairly succinct way. Overall I believe the authors arguments are valid and of interest to the community, even if they are not necessarily new. However there are several important instances with questionable gaps in the logic and some critical parts of the method description

seem to be missing.

An immediate question I had when reading the manuscript is that the term "expert knowledge" does not appear to be defined / explained. Should it be contrasted to some some other kind of "general" knowledge (or "alternative" knowledge?). As a result it is unclear what type of information/data do the authors consider to be "expert knowledge", and which they do not, and the discussion that follows then becomes a bit too vague. For example, is the classification into top-down vs bottom-up considered expert knowledge? I would have argued this would nowdays represent "general" knowledge rather than "expert" knowledge, and so forth.

The case study description should also more clearly categorise and describe the types of "expert knowledge" used. Some of the expert knowledge seems to be just a particular data set (which is in principle accessible to "non-expert" modellers also). Perhaps I missed something here, in which case a clarification in the manuscript is still needed to prevent such confusion for other readers.

The manuscript makes an overly general statement is when suggesting "modellers, or 'dry' hydrologists" tend to develop theories at the catchment scale. Surely this would not apply to hydrologists that use "physically-based" models, which are classic bottom-up approaches. And the contrast to "experimentalists, or 'wet' hydrologists" is itself questionable - surely some of them work at the small scale and others extrapolate to the larger scale. Some additional supporting references here are needed. Or maybe clarify that "modellers" here mean "conceptual" modellers (?)

The ANOVA analysis used to estimate dominant source of uncertainty is a worthwhile undertaking, but the description here lacks essential detail. For example, how are the individual terms in equation 6 calculated? Clearly some assumptions need to be made here, eg, about the errors in the rainfall inputs (eg, Renard et al WRR2011), about the errors in the maps, and so on - how are the assumptions made and how are they checked in this study? Otherwise the conclusions of this analysis would not really be

justified.

Pages 12 and 16: "Model realism" ... Of course every scientist and engineer would want their model to be "realistic", and there have been many opinion papers on this in hydrology advocating improving the "realism", whatever that means beyond ultimately matching some performance criteria. In this manuscript this issue is raised on Page 12, but then after a fairly basic modelling attempt produced results deemed "unrealistic" (page 13, lines 10-20), any improvement was deemed "beyond the scope". So I am not sure the manuscript and its case study in their current form can convincingly call on other modellers to achieve such "realism".

Other

1. Page 2 Line 5: What does the "It" refer to? "Expert knowledge"? Or "linking observations and laws"? The latter would make far more sense, but is not apparent from the way the sentence is written.

2. Page 2 Line 24 "Wet and dry" - this usage here is confusing and unnecessary. Taken literally "dry hydrologists" might be mistaken for hydrologists working in dry (arid/semi-arid) catchments. I think it is fun to refer to experimentalists and modellers as wet and dry, but I think once the point is made to continue using these terms is not necessary and can lead to confusion.

3. Page 3 line 19 "too simplistic" - I think this requires some statement of the purpose - too simplistic to achieve what?

4. Pages 3-4: Fenicia et al WRR2016 is an example where bottom-up ("distributed" and scaled up) and "top-down" (conceptual) approaches (as per definitions given earlier) are combined, applied and several hypotheses about process representations and hydrological controls were tested.

5. Page 5 line 25: This text correctly refers to the distinction between top-down and bottom-up approaches not being sharp - this aspect was overlooked in the earlier

pages when discussing these two approaches, which lead to some too-strong statements being made.

6. Page 6 Line 25: "parameter allocation" sounds a bit awkward? Wouldn't "estimation" be a better choice here, covering all options (though measurement, calibration, etc)?

7. Page 6 line 31: give a reference for this "so-called hydrological approach", and explain how it is used in this work

8. Page 7 line 22: "storage parameter" might read better here than "storage constant"?

9. Eqn 1: numbering each of these 2 equations individually would be helpful

10. Page 8 line 20: what is a "modelling chain combination"? please briefly define/explain what this means

11. Page 9 eqn 5 - clarify that "KGE applied to runoff time series" or similar, for clarity

12. Page 9 line 18-23: Please define these P-factors and R-factors by equations, at least to the same standard as the KGE in eqn 1. Then all metrics become properly documented and easily reproducible without guessing how exactly they are calculated.

13. Section 4.5 It is good that the study is noting its limitations. I would also suggest the overall calibration approach used here is quite simplistic - it is quite possible a more detailed application would've produced quite different results. Worth also noting the use of a single case study makes it impossible to know if similar conclusions would hold elsewhere. This does not invalidate the results, just should be noted as one of the limitations.

14. Section 5 Conclusions - to make sense of this, need to list the actual "expert knowledge" used here. This relates to a major concerned raised earlier that this concept is not sufficiently clearly delineated here.

15. Fig 1 - I found this figure rather confusing - somehow physically based models became closer to experimentalists approaches than conceptual models? I would not

have said that, these kinds of models are certainly not developed by "field" "wet" hydrologists ... I think they do not fit within this "planar" dichotomous figure, they almost form another dimension of their own.

16. Fig 4 - worth providing references to several earlier studies where such "distributed model structures" were used.

17. Fig 6 - what are the benchmark options for the process maps, parameterizations, and parameter values?

18. Figure 7 - not sure I am seeing in the figure how you arrived at the last sentence in the caption.

---

## Referee Comment (RC2) · U. Ehret (Referee) · 26 Jul 2017

**Review of Manuscript**

**'How can expert knowledge increase the realism of conceptual hydrological models? A case study in the Swiss Pre-Alps '**

**by M. Antonetti and M. Zappa**

Dear Editor, dear Authors,

I have reviewed the aforementioned work. My conclusions and comments are as follows:

1. **Scope**

The article is within the scope of HESS.

2. **Summary**

The authors evaluate the influence of several choices in an event-based, conceptual hydrological modeling exercise in mesoscale (up to 445 km²) catchments. This includes choices on rainfall input, initial conditions, dominant runoff process mapping and model parameterization strategy. For the former three, the choices can be ranked according to an expert-based, prior assumption of 'quality', the latter distinguishes a 'bottom-up' and a 'top-down' approach. Comparison of rainfall-runoff simulations of many combined realizations of the above choices with discharge observations reveal that i) the most complex (assumed most informative) process maps did only slightly outperform the simpler versions, ii) the bottom-up and top-down approaches differed only for short events, slightly in favor of the bottom-up approach, iii) the effect of forcing quality was hard to interpret due to compensation effects and iv) the top-down approach seems to be more robust and bottom-up tends to overconfidence.

3. **Overall ranking**

The work is ranked 'Minor revision'.

4. **Evaluation**

The study is well-designed, clearly reported, the conclusions are supported by the data and results. So just a few comments from my side:

- Title: The study has a strong focus on the concept of dominant runoff processes; in fact all of the experiments include DRP –based choices. This should be reflected in the title. E.g. 'How can expert knowledge increase the realism of conceptual hydrological models? A case study based on the concept of dominant runoff processes in the Swiss Alps.

- P3L3: applied instead of solved (or 'used as constraints')

- P4L31 pp: The connection between the text, Table 2 and Figure 3 should be improved:
  - Reverse the order of a), b) and c) in Fig 3 to match the order of Table 2
  - In Fig 3, add some textual info of the meaning of the RT's (e.g. column 3 of Table 2)
  - Add some more explanation about the maps in Fig 3 to the text (e.g. what is the raw resolution of the maps)

- P5L20 and P25 Table 2: It is not entirely clear to me how the mapping from 9 → 5 and 12 → 5 types was done.

- P6 section 2.2.1: So PREVAH was used to initialize SSM. How was SUZ initialized?

- P7 section 2.3.1: How was GS1H determined?

- P7L12: Can you explain in more detail the optimization against generalized response curves?

- P7 section 2.3.2: How was the routing parameter chosen for the top-down approach? Same as for bottom-up?
- P11L6: overestimate instead of underestimate
- P11L23: Can you explain 'interaction' in this context?

Yours sincerely,

Uwe Ehret

---

## Author Comment (AC2) · 31 Aug 2017

**The study is well-designed, clearly reported, the conclusions are supported by the data and results. So just a few comments from my side:**

Reply: We would like to thank Uwe Ehret for his suggestions and are happy to read his positive comments on our manuscript. In the following, we replied to each point raised by the reviewer and indicated how we will implement his suggestions in the revised manuscript.

**Title: The study has a strong focus on the concept of dominant runoff processes; in fact all of the experiments include DRP–based choices. This should be reflected in the title. E.g. 'How can expert knowledge increase the realism of conceptual hydrological models? A case study based on the concept of dominant runoff processes in the Swiss Alps.**

Reply: Thanks for this suggestion, which will be embraced in the revised manuscript.

**P3L3: applied instead of solved (or 'used as constraints')**

Reply: Agreed.

**P4L31 pp: The connection between the text, Table 2 and Figure 3 should be improved:**

**Reverse the order of a), b) and c) in Fig 3 to match the order of Table 2**

**In Fig 3, add some textual info of the meaning of the RT's (e.g. column 3 of Table 2)**

**Add some more explanation about the maps in Fig 3 to the text (e.g. what is the raw resolution of the maps)**

Reply: Agreed. In the revised manuscript we will reverse the order of the process maps in Fig. 3, extend its legend and integrate more information in the caption.

**P5L20 and P25 Table 2: It is not entirely clear to me how the mapping from 9 → 5 and 12 → 5 types was done.**

Reply: The nine DRP classes (HOF1-2,SOF1-3,SSF1-3, and DP) of the mapping approach after Müller et al. (2009) are reclassified in five runoff types (RT1-5) according to Table 2. In addition to the same 9 DRP classes used by Müller et al. (2009), the original method of Schmocker-Fackel et al. (2007) allows areas where water is artificially drained (D1-3) to be identified, provided that maps of tile drain systems are available. As these were not available for our study catchments, the original 12 classes get reduced to nine, and the same reclassification criteria as for Müller et al.'s (2009) approach were used (Table 2). we will add this information in the revised manuscript.

**P6 section 2.2.1: So PREVAH was used to initialize SSM. How was SUZ initialized?**

Reply: Each simulation was started (at least one day before the beginning of the rainfall event and) sufficiently far from possible previous events, so that it was possible to assume that no overland flow and no subsurface flow was occurring in the first time step. Consequently, SUZ was set equal to 0. We will add this in the revised manuscript.

**P7 section 2.3.1: How was GS1H determined?**

Reply: GS1H was determined based on expert knowledge. Considered the size (from 0.5 $km^2$ up to 2 $km^2$) of the subcatchments, into which the main catchment was subdivided, an initial range between 1h and 3h was considered to be plausible for the storage constant governing the concentration of subsurface flow. The same initial ranges were also used in Antonetti et al. (2017). We will therefore add this reference at the end of the sentence in question.

**P7L12: Can you explain in more detail the optimization against generalized response curves?**

Reply: Based on the results of the sprinkling experiments, on physical properties of soils, and on the field expertise of the authors who mapped the runoff types in the investigated areas, plausible value ranges were defined a priori for each parameter of the bottom-up setup (Table 4 of Antonetti et al., 2017). Contextually, idealised response curves were defined for each runoff type (Figure 5 of Antonetti et al., 2017). These curves are idealised results from the sprinkling experiments and represent the expected behaviour of the correspondent runoff type in terms of intensity to runoff contribution. The initial ranges of each model parameter were defined a priori for each runoff type, according to the characteristics of the DRPs belonging to it (Table 3 of Antonetti et al., 2017). With regard to the partitioning of runoff within those runoff types, where different DRPs can occur (e.g. runoff type 2, where both SOF2 and SSF1 can take place), the parameter ranges were defined in a manner that allows equifinal combinations to be considered (Beven, 2006). As a result, overland flow and subsurface flow can be partitioned in different ways, provided that the total contribution to runoff reflects that of the correspondent response curve. These ranges were then optimised against the characteristic response curve of each runoff type by considering only the 1% best runs of a Monte Carlo simulation with 10'000 runs. Since response curves instead of hydrographs are used for the optimisation, the root mean square error (RMSE) was used as objective function.

In the revised manuscript we will extend the description of the optimisation against the generalised response curves. We will try however to keep the explanation as short as possible by referring the reader to our previous study (Antonetti et al. 2017).

**P7 section 2.3.2: How was the routing parameter chosen for the top-down approach? Same as for bottom-up?**

Reply: For the bottom-up setup the runoff routing in the channel was calculated with the hydraulic approach after Schulla (1997), i.e. by considering the flow times calculated with a Strickler coefficient of 30 $m^{1/3}$ $s^{-1}$. For the top-down setup no explicit routing parameter is defined, as routing is coupled with runoff generation and concentration (as is done currently in PREVAH). Therefore, the storage constants K0Hi and K1Hi (i = 1-5) account also for routing. Their values are allocated based on the strategy after Gharari et al. (2014), which is described in section 2.3.2 of the manuscript.

**P11L6: overestimate instead of underestimate**

Reply: Of course! We really appreciate the attention the referee put in reviewing our manuscript.

**P11L23: Can you explain 'interaction' in this context?**

Reply: Within the ANOVA framework, an interaction is linked with the concept of covariance and is defined as the effect of a factor that depend on the effects of one or more other factors (see for instance Köplin et al., 2013). We will specify this in the revised manuscript.

References

Antonetti, M., Scherrer, S., Kienzler, P. M., Margreth, M. and Zappa, M.: Process-based Hydrological Modelling: The Potential of a Bottom-Up Approach for Runoff Predictions in Ungauged Catchments, Hydrol. Process., doi:10.1002/hyp.11232, 2017.

Beven, K.: A manifesto for the equifinality thesis, in Journal of Hydrology, vol. 320, pp. 18–36., 2006.

Gharari, S., Hrachowitz, M., Fenicia, F., Gao, H. and Savenije, H. H. G.: Using expert knowledge to increase realism in environmental system models can dramatically reduce the need for calibration, Hydrol. Earth Syst. Sci., 18(12), 4839–4859, doi:10.5194/hess-18-4839-2014, 2014.

Köplin, N., Schädler, B., Viviroli, D. and Weingartner, R.: The importance of glacier and forest change in hydrological climate-impact studies, Hydrol. Earth Syst. Sci., 17(2), 619–635, doi:10.5194/hess-17-619-2013, 2013.

Schulla, J.: Hydrologische Modellierung von Flussgebieten zur Abschätzung der Folgen von Klimaänderungen., 1997.

---

## Author Response (AR1)

**The manuscript discusses the role of expert knowledge in hydrological modelling, contrasts several perspectives described as "modeller" and "experimentalist", and provides a simple case study illustrating the discussion.**

**I generally enjoyed reading the manuscript, it is written in an easy-to-follow style, and does not oversell its results. The case study is presented in a fairly succinct way. Overall I believe the authors arguments are valid and of interest to the community, even if they are not necessarily new. However there are several important instances with questionable gaps in the logic and some critical parts of the method description seem to be missing.**

Reply: We would like to thank the anonymous reviewer for his comments and suggestions. His/Her comments regard mainly the introductory and methodological part of the manuscript, whereas no changes to the results and discussion part were requested. We acknowledge that some of our arguments might not be seen as new and may already have been enunciated by other authors. However, we found exclusively review papers (e.g. Clark et al., 2017) or opinion papers (e.g. Hrachowitz and Clark, 2017) on the topic. Our contribution is therefore intended to be one of the first hands-on exercise on the coupling of bottom-up and top-down thinking, with a specific regard to DRP-based modelling approaches. In the following, we replied to each point raised by the reviewer and indicated how we implemented the suggestions in the revised manuscript.

**An immediate question I had when reading the manuscript is that the term "expert knowledge" does not appear to be defined / explained. Should it be contrasted to some other kind of "general" knowledge (or "alternative" knowledge?). As a result it is unclear what type of information/data do the authors consider to be "expert knowledge", and which they do not, and the discussion that follows then becomes a bit too vague. For example, is the classification into top-down vs bottom-up considered expert knowledge? I would have argued this would nowdays represent "general" knowledge rather than "expert" knowledge, and so forth.**

Reply: With the term "expert knowledge" we refer to someone's acquaintance with hydrological sciences as a result of study and experience. In this context, the term "expert" should be understood as "relating to a person who has knowledge in a particular field" (WordReference Random House Learner's Dictionary of American English © 2017) and it should not be contrasted to other kinds of knowledge. The expression "expert knowledge" was already used by several authors: Gharari et al. (2014), Hrachowitz et al. (2014), and Safavi et al. (2015) used it in the title, whereas e.g. Antonetti et al. (2016), Bahremand (2016), Hellebrand et al. (2011), Hümann and Müller (2013), Nijzink et al. (2016), Peschke et al. (1999), Sivakumar (2004) and Smith et al. (2013) refer to it in their manuscript. In the revised manuscript, we will therefore keep the expression as it is for consistency.

As acquaintance is uncountable by definition, information to be considered as "expert knowledge" cannot be listed. Of course, the expert knowledge of a hydrologist with 30 years of experience cannot be compared with that of a PhD student. In the same way, expert knowledge of a modeller will differ from that of an experimentalist. What we tried to do in this study is to compare *different ways* of implementing expert knowledge in hydrological conceptual models, following either a top-down or a bottom-up thinking. In the revised manuscript we therefore added a definition of „expert knowledge" at the beginning of the

introductory section and explicitly referred to the different ways (top-down or bottom-up) for the use of expert knowledge in the process of scientific reasoning.

**The case study description should also more clearly categorise and describe the types of "expert knowledge" used. Some of the expert knowledge seems to be just a particular data set (which is in principle accessible to "non-expert" modellers also). Perhaps I missed something here, in which case a clarification in the manuscript is still needed to prevent such confusion for other readers.**

Reply: As mentioned in the previous reply, we would not refer to *types* of expert knowledge but rather to *ways* of using it during the three steps of the process-based modelling step (i.e. process mapping, process parameterisation, and parameter allocation). With regard to process mapping, the mapping approach of Schmocker-Fackel et al. (2007) and Margreth et al. (2010) is the one involving the largest amount of expert knowledge, which is used for manually mapping the small test areas (pg. 5, line 18 of the revised manuscript) and setting-up the mapping algorithm. In contrast, the two other mapping approaches used for the study - Gharari et al.'s (2011) approach based on topographical information only, and Müller et al.'s (2009) approach based on DTM, geological and land use maps - barely rely on expert knowledge, which was used by the corresponding authors exclusively for verifying the classification criteria. One of the research questions addressed in the manuscript is therefore whether the use of more expert knowledge during the mapping phase improves hydrological simulations given unavoidable uncertainties in the forcing data (pg. 4, line 10).

We added a definition of expert knowledge (see first reply) and we emphasised on the ways of using it (pg. 2, line 8 and 26; pg. 4, line 10). We hope this will prevent confusion for the reader.

**The manuscript makes an overly general statement is when suggesting "modellers, or 'dry' hydrologists" tend to develop theories at the catchment scale. Surely this would not apply to hydrologists that use "physically-based" models, which are classic bottom-up approaches. And the contrast to "experimentalists, or 'wet' hydrologists" is itself questionable - surely some of them work at the small scale and others extrapolate to the larger scale. Some additional supporting references here are needed. Or maybe clarify that "modellers" here mean "conceptual" modellers (?)**

Reply: To our knowledge, the two appellations *modellers* and *experimentalists* were introduced by Seibert and McDonnell (2002) in their study on the use of soft data during model building and calibration. At the end of the paragraph in question we therefore cited their work. Some modellers will certainly develop physically-based models, as well as some experimentalists will focus on the regionalisation of small scale outcomes. This was discussed in the next paragraph, where we clearly state that the two philosophies (bottom-up and top-down) can be interpreted differently when applied to hydrological modelling (page 3, line 3-11 of the revised manuscript). As suggested by the reviewer, we added the term "conceptual" before "modeller" to prevent confusion to the reader.

**The ANOVA analysis used to estimate dominant source of uncertainty is a worthwhile undertaking, but the description here lacks essential detail. For example, how are the individual terms in equation 6 calculated? Clearly some assumptions need to be made here, eg, about the errors in the rainfall inputs (eg, Renard et al WRR2011), about the**

**errors in the maps, and so on - how are the assumptions made and how are they checked in this study? Otherwise the conclusions of this analysis would not really be justified.**

Reply: With the analysis of variance (ANOVA) it is possible to quantify the relative impact of source of uncertainty, named factors, on a given response variable, in our case the Kling-Gupta-Efficiency, based on the assumption that the output variance can describe the uncertainty of an environmental system (von Storch and Zwiers, 1999). In addition, the approach assumes independence between the different levels (e.g., between the different process maps). This assumption is not fulfilled, as e.g. three datasets use the same forcing (i.e. Combiprecip), and the two model setups are based on the same runoff generation module (i.e. RGM-PRO).

The method is clearly less complex than the Bayesian total error analysis (BATEA) method suggested by the reviewer, and others, but has advantages in terms of robustness (Tang et al., 2007), and is easy to implement thanks to the R package *stats* (version 3.2.2; R Core Team) where a function for the ANOVA is built in.

For the calculation of the individual terms in equation 6 we refer to equations from 18 to 22 of Bosshard et al. (2013). However, we believe that a longer description of the method would distract the reader from the objective of the study. It should be also noticed that the length and the amount of information contained in the paragraph where ANOVA is described, is comparable to that of other studies relying on ANOVA for quantifying uncertainty sources (e.g. Addor et al., 2014; Finger et al., 2012; Köplin et al., 2013; Rössler et al., 2012).

In the revised manuscript we therefore added:

- A reference to a comprehensive description of the ANOVA method, e.g. von Storch and Zwiers (1999)
- A sentence about the assumption of independency between different levels of a given factor.
- A sentence on how to calculate each individual term of equation (6) and refer to Bosshard et al. (2013) for further details.

**Pages 12 and 16: "Model realism" ... Of course every scientist and engineer would want their model to be "realistic", and there have been many opinion papers on this in hydrology advocating improving the "realism", whatever that means beyond ultimately matching some performance criteria. In this manuscript this issue is raised on Page 12, but then after a fairly basic modelling attempt produced results deemed "unrealistic" (page 13, lines 10-20), any improvement was deemed "beyond the scope". So I am not sure the manuscript and its case study in their current form can convincingly call on other modellers to achieve such "realism".**

Reply: Here we disagree with the reviewers' definition of the expression "improving the realism". In our view, it is not exclusively a question of matching performance criteria, but more an attempt of letting the model behave according to the hydrologist perception of how it should behave. The crucial point here is the perception of *which* hydrologist should we refer to. For instance, Gharari et al. (2014b) claimed that "using expert knowledge to increase realism in environmental system models can dramatically reduce the need for calibration". In a parallel study, they define expert knowledge as "the *modeler's* perception of catchment behaviour and characteristics" (Gharari et al., 2014a). According to the two hydrologists

types described in this study, the reviewer will agree that the *experimentalist's* perception would be more detailed than the modeller's one.

As we exclusively compared existing methods for process mapping, process parameterisation and parameter allocation, any improvement of one of those methods was actually "beyond the scope" of the study. However, we disagree with the reviewer's reconstruction of our reasoning in section 4.1, as, for instance, the expression "beyond the scope"(pg. 13 line 20) belongs to the next section of the manuscript (4.2), which is not about model realism. The line of reasoning of section 4.1. is therefore following:

1. A modeller might conclude that using one of the two top-down mapping approaches (MU09 and GH11) used here may be the best choice in terms of cost-benefit, as the results obtained with them were, on average, only slightly lower than those obtained with the process maps with higher involvement of expert knowledge (SF07).
2. This conclusion is however not acceptable from an experimentalist point of view. The results may seem acceptable at the gauging stations, but the local representation of the DRP mapped would most likely differ from that expected by an experimentalist, as shown by Antonetti et al. (2016).
3. Therefore, we advocate that the hydrological community should aspire to develop models able to reproduce processes in a "realistic" way (i.e. in agreement with the *experimentalists'* expectation) at least at the sub-catchment or, even better, at the hillslope scale. At these scales, the added value of the process maps with higher involvement of expert knowledge should get highlighted.
4. As runoff measurements at these scales were not available on the investigated catchments, this issue remains a speculation, and will be verified during future research.

In the original manuscript we made a call based on our opinion (page 13, line 30) for the use of experimentalist's rather than modeller's perception for verification purposes (page 12, line 32). In addition, we stated that our considerations about the scales, at which model realism should be achieved, will be verified during future research (page 14, line 4).

**Other**

**1. Page 2 Line 5: What does the "It" refer to? "Expert knowledge"? Or "linking observations and laws"? The latter would make far more sense, but is not apparent from the way the sentence is written.**

Reply: "It" is referred to the process of "linking observations and laws". In the revised manuscript we rephrased this sentence.

**2. Page 2 Line 24 "Wet and dry" - this usage here is confusing and unnecessary. Taken literally "dry hydrologists" might be mistaken for hydrologists working in dry (arid/semiarid) catchments. I think it is fun to refer to experimentalists and modellers as wet and dry, but I think once the point is made to continue using these terms is not necessary and can lead to confusion.**

Reply: Agreed. In the revised manuscript we changed "wet and dry" with their corresponding appellations "experimentalists and modellers".

**3. Page 3 line 19 "too simplistic" - I think this requires some statement of the purpose - too simplistic to achieve what?**

Reply: Agreed. In the revised manuscript we modified the sentence as follows: "Top-down models and parameterisations may be too simplistic to depict the spatial variability of runoff processes within a given catchment and, therefore, require calibration […]"

**4. Pages 3-4: Fenicia et al WRR2016 is an example where bottom-up ("distributed" and scaled up) and "top-down" (conceptual) approaches (as per definitions given earlier) are combined, applied and several hypotheses about process representations and hydrological controls were tested.**

Reply: We want to thank the anonymous reviewer for the suggested reference, which was integrated in the revised manuscript.

**5. Page 5 line 25: This text correctly refers to the distinction between top-down and bottom-up approaches not being sharp - this aspect was overlooked in the earlier pages when discussing these two approaches, which lead to some too-strong statements being made.**

Reply: Here we disagree with the reviewer, since we stated at page 3, line 1 of the original manuscript that " Different interpretations of the two philosophies have been applied in hydrological modelling." In the revised manuscript we tried to state this more clearly: " The distinction between top-down and bottom-up is not sharp and different interpretations of the two philosophies have been applied in hydrological modelling."

**6. Page 6 Line 25: "parameter allocation" sounds a bit awkward? Wouldn't "estimation" be a better choice here, covering all options (though measurement, calibration, etc)?**

Reply: This issue was already raised by a reviewer in a previous publication (Antonetti et al., 2017), and the use of the expression "parameter allocation" was suggested by the other reviewer. We would therefore like to keep the expression "parameter allocation" throughout the manuscript, for consistency with Antonetti et al. (2017). In his recent opinion paper, Bahremand (2016) defined "parameter allocation" as "a logic-based specification", which completely fits in the context of the present study.

**7. Page 6 line 31: give a reference for this "so-called hydrological approach", and explain how it is used in this work.**

Reply: In the revised manuscript we added the reference Dyck and Peschke (1995) but the explanation of how the hydrological approach is used in this work is already given in pg 7 line 20-25 of the revised manuscript; there, we explicitly referred to the hydrological approach in the revised manuscript.

**8. Page 7 line 22: "storage parameter" might read better here than "storage constant"?**

Page: The reviewer might be right maintaining that "parameter" would be better than "constant" as a worth choice. However, if the reviewer agrees, we would like to keep the expression "storage constant" for consistency with Antonetti et al. (2016, 2017).

**9. Eqn 1: numbering each of these 2 equations individually would be helpful.**

Reply: Agreed.

**10. Page 8 line 20: what is a "modelling chain combination"? please briefly define/explain what this means.**

Reply: The term in question is borrowed from the HEPEX (Hydrologic Ensemble Prediction EXperiment) community (https://hepex.irstea.fr/), where it is usually used to indicate a cascade of modelling elements, such as forcing data, hydrological models, pre- or post-processing methods etc. (see for instance Bosshard et al., 2013). In this work, a single modelling chain consists of a given dataset of forcing data, a DRP map, and a model setup.

In the revised manuscript we added a brief explanation.

**11. Page 9 eqn 5 - clarify that "KGE applied to runoff time series" or similar, for clarity**

Reply: Agreed. In the revised manuscript we modified the sentence as follows: "*Runoff* simulations were evaluated with the Kling Gupta Efficiency (KGE; Gupta et al., 2009).

**12. Page 9 line 18-23: Please define these P-factors and R-factors by equations, at least to the same standard as the KGE in eqn 1. Then all metrics become properly documented and easily reproducible without guessing how exactly they are calculated.**

Reply: Agreed. In the revised manuscript we reported the equations for P- and R-factors.

**13. Section 4.5 It is good that the study is noting its limitations. I would also suggest the overall calibration approach used here is quite simplistic - it is quite possible a more detailed application would've produced quite different results. Worth also noting the use of a single case study makes it impossible to know if similar conclusions would hold elsewhere. This does not invalidate the results, just should be noted as one of the limitations.**

Reply: Here we are not really sure which calibration criteria the reviewer is referring to, as both top-down and bottom-up model setups were applied uncalibrated on the study catchments.

The fact that a single case study makes it impossible to know if similar conclusions would hold elsewhere was already pointed out in pg. 16 lines 4-5: "Finally, to generalise the findings of this study, the number of catchments and events investigated should be increased considerably."

**14. Section 5 Conclusions - to make sense of this, need to list the actual "expert knowledge" used here. This relates to a major concerned raised earlier that this concept is not sufficiently clearly delineated here.**

Reply: See reply to the major concern.

**15. Fig 1 - I found this figure rather confusing - somehow physically based models became closer to experimentalists approaches than conceptual models? I would not have said that, these kinds of models are certainly not developed by "field" "wet" hydrologists ... I think they do not fit within this "planar" dichotomous figure, they almost form another dimension of their own.**

Reply: Agreed. We removed the terms "modellers" and "experimentalists" from Fig 1 in the revised manuscript.

**16. Fig 4 - worth providing references to several earlier studies where such "distributed model structures" were used.**

Reply: Agreed.

**17. Fig 6 - what are the benchmark options for the process maps, parameterizations, and parameter values?**

Reply: There is no benchmark for process maps and model setup, as they are all depicted in Fig. 8 and Fig. 10. The reason why that forcing data was used as benchmark is reported on pg.9 line 31-32 of the revised manuscript.

**18. Figure 7 - not sure I am seeing in the figure how you arrived at the last sentence in the caption.**

In the revised manuscript we modified the last sentence of the caption as follows: "The SF07 map reproduced best the peak runoff with the bottom-up setup, whereas the GH11 map outperformed the other maps with the top-down setup".


**The study is well-designed, clearly reported, the conclusions are supported by the data and results. So just a few comments from my side:**

Reply: We would like to thank Uwe Ehret for his suggestions and are happy to read his positive comments on our manuscript. We acknowledge that the reviewer's comments concern exclusively the methodological part of the manuscript, whereas no changes to the results and discussion part were requested. In the following, we replied to each point raised by the reviewer and indicated how we implemented his suggestions in the revised manuscript.

**Title: The study has a strong focus on the concept of dominant runoff processes; in fact all of the experiments include DRP–based choices. This should be reflected in the title. E.g. 'How can expert knowledge increase the realism of conceptual hydrological models? A case study based on the concept of dominant runoff processes in the Swiss Alps.**

Reply: Thanks for this suggestion, which was embraced in the revised manuscript.

**P3L3: applied instead of solved (or 'used as constraints')**

Reply: Agreed.

**P4L31 pp: The connection between the text, Table 2 and Figure 3 should be improved: Reverse the order of a), b) and c) in Fig 3 to match the order of Table 2**
**In Fig 3, add some textual info of the meaning of the RT's (e.g. column 3 of Table 2)**
**Add some more explanation about the maps in Fig 3 to the text (e.g. what is the raw resolution of the maps)**

Reply: Agreed. In the revised manuscript we reversed the order of the process maps in Fig. 3, extended its legend and integrated more information in the text.

**P5L20 and P25 Table 2: It is not entirely clear to me how the mapping from 9 → 5 and 12 → 5 types was done.**

Reply: The nine DRP classes (HOF1-2, SOF1-3, SSF1-3, and DP) of the mapping approach after Müller et al. (2009) are reclassified in five runoff types (RT1-5) according to Table 2. In addition to the same 9 DRP classes used by Müller et al. (2009), the original method of Schmocker-Fackel et al. (2007) allows areas where water is artificially drained (D1-3) to be identified, provided that maps of tile drain systems are available. As these were not available for our study catchments, the original 12 classes get reduced to nine, and the same reclassification criteria as for Müller et al.'s (2009) approach were used (Table 2). We added this information in the revised manuscript.

**P6 section 2.2.1: So PREVAH was used to initialize SSM. How was SUZ initialized?**

Reply: Each simulation was started at least one day before the beginning of the rainfall event and sufficiently far from possible previous events, so that it was possible to assume that no overland flow and no subsurface flow was occurring in the first time step. Consequently, SUZ was set equal to 0. We added this in the revised manuscript.

**P7 section 2.3.1: How was GS1H determined?**

Reply: GS1H was determined based on expert knowledge. Considered the size (from 0.5 km$^2$ up to 2 km$^2$) of the sub-catchments, into which the main catchment was subdivided, an initial range between 1h and 3h was considered to be plausible for the storage constant governing the concentration of subsurface flow. The same initial ranges were also used in Antonetti et al. (2017). We therefore added this reference at the end of the sentence in question.

**P7L12: Can you explain in more detail the optimization against generalized response curves?**

Reply: Based on the results of the sprinkling experiments, on physical properties of soils, and on the field expertise of the authors who mapped the runoff types in the investigated areas, plausible value ranges were defined a priori for each parameter of the bottom-up setup (Table 4 of Antonetti et al., 2017). Contextually, idealised response curves were defined for each runoff type (Figure 5 of Antonetti et al., 2017). These curves are idealised results from the sprinkling experiments and represent the expected behaviour of the correspondent runoff type in terms of intensity to runoff contribution. The initial ranges of each model parameter were defined a priori for each runoff type, according to the characteristics of the DRPs belonging to it (Table 3 of Antonetti et al., 2017). With regard to the partitioning of runoff within those runoff types, where different DRPs can occur (e.g. runoff type 2, where both SOF2 and SSF1 can take place), the parameter ranges were defined in a manner that allows equifinal combinations to be considered (Beven, 2006). As a result, overland flow and subsurface flow can be partitioned in different ways, provided that the total contribution to runoff reflects that of the correspondent response curve. These ranges were then optimised against the characteristic response curve of each runoff type by considering only the 1% best runs of a Monte Carlo simulation with 10'000 runs. Since response curves instead of hydrographs are used for the optimisation, the root mean square error (RMSE) was used as objective function.

In the revised manuscript we extended the description of the optimisation against the generalised response curves. We tried however to keep the explanation as short as possible by referring the reader to our previous study (Antonetti et al. 2017).

**P7 section 2.3.2: How was the routing parameter chosen for the top-down approach? Same as for bottom-up?**

Reply: For the bottom-up setup the runoff routing in the channel was calculated with the hydraulic approach after Schulla (1997), i.e. by considering the flow times calculated with a Strickler coefficient of 30 $m^{1/3}$ $s^{-1}$. For the top-down setup no explicit routing parameter is defined, as routing is coupled with runoff generation and concentration (as is done currently in PREVAH). Therefore, the storage constants K0Hi and K1Hi (i = 1-5) account also for routing. Their values are allocated based on the strategy after Gharari et al. (2014), which is described in section 2.3.2 of the manuscript.

**P11L6: overestimate instead of underestimate**

Reply: Of course! We really appreciate the attention the referee put in reviewing our manuscript.

**P11L23: Can you explain 'interaction' in this context?**

Reply: Within the ANOVA framework, an interaction is linked with the concept of covariance and is defined as the effect of a factor that depends on the effects of one or more other factors (see for instance Köplin et al., 2013). We specified this in the revised manuscript.


Authors' response to reviewer nr. 3

**The article questions the added value of expert knowledge in the simulation of the rainfall-streamflow relationship by analyzing the sensitivity of a process-oriented model to the mapping of dominant runoff processes. The analysis is made on four nested catchments in the Swiss Alps.**

**I found this is an interesting article, which is clear, easy to follow and well written. I have some requests for a few clarifications and changes in the text. I advise publication after minor modifications.**

Reply: We would like to thank the anonymous reviewer for his/her comments and suggestions. We acknowledge that the reviewer's comments concern mainly the introductory and methodological part of the manuscript, whereas the only concern about the results is the small number of events investigated, which could prevent our conclusions to be generalised. Of course, we are aware of this limitation, and explicitly referred to it in the title (this is a "case study") and in the limitation section. An increase of the simulated events was therefore unfeasible due to time restrictions, but we encourage future research to consolidate our findings.

In the following, we replied to each point raised by the reviewer and indicated how we implemented some suggestions or why we did not embrace some others in the revised manuscript.

**Specific comments:**

**1.     Introduction: The authors much emphasize the issue of spatial process mapping. They do not mention much the issue of time scale of these processes. Are the time scales of dominant processes well represented in the model? Though this can be a cause of model failure, this is often a neglected problem. I know this is not the main focus of the article, but this issue of time appear several times in the article (e.g. P5, L5). The literature on this may be shortly discussed somewhere in the article, in the introduction or the discussion (see e.g. work by Savenije and others).**

Reply: Unfortunately, we did not fully understand what the reviewer means with "time scale issues", and the reference provided is too vague for us to elaborate on this topic.

The issue of spatial process mapping is just one of the three steps for flood predictions we addressed in the introduction (see Fig.1 of the manuscript). For each of these steps, we reported both bottom-up and top-down approaches to handle them.

If the reviewer refers to whether the model is able to reproduce both fast processes as Hortonian Overland Flow (HOF) and delayed processes as e.g. Deep Percolation (DP) RGM-PRO we refer to a previous publication (Antonetti et al., 2017). In that study, RGM-PRO was:

- successfully applied on hillslopes with a simulation timestep and forcing data with a resolution of 10 min (Antonetti et al., 2017).
- successfully upscaled to the regional scale with forcing data with hourly resolution but with an internal integration timestep of 10 minutes, to prevent numerical problems (Kavetski and Clark, 2010).

As the issue of time scale is not addressed in the paper, we believe that discussing it in the introduction could be misleading for the reader. We already mentioned in the limitation section that we focussed on the event scale, as the process maps used are focussed on floods and RGM-PRO is an event-based model. We also mentioned that investigating the influence of different levels of expert knowledge on processes other than runoff generation (e.g. evapotranspiration and interception), which involve other (i.e. longer) temporal scales, could be a possible direction for future research.

**2.     P4,L22: Maybe show these administrative limits in the first map of Fig. 2 (if it does not make the map too messy) .**

Reply: We tried to add the canton borders to the first map of Fig. 2. However, it resulted in a map with too high amount of information. We would therefore like to keep the figure as it is.

**3.     P5,L11-17: This part may be presented with bullet points, to ease reading.**

Reply: Agreed.

**4.     P5,L19: "SF07" is used before it is defined.**

Reply: We introduced earlier the abbreviation in the revised manuscript.

**5.     P6,L3-8: Some parameters do not appear in Fig. 4 and could be added to better follow the model description.**

Reply: The parameter BETA, which controls the separation of rainfall between the storage of plant-available soil moisture (SSM) and the runoff generation module (SUZ). As BETA was fixed to a value of 3 accordingly to a previous study (Antonetti et al., 2016), we removed it from Fig. 4 in a first place. In the revised manuscript we added BETA in Fig. 4 as well as Table 3.

**6.     P6,L10-13: Is this study made in a forecasting context? If yes, this could be mentioned since it may have an importance in the results, given the role of data assimilation.**

Reply: The main focus of this study is on the use of expert knowledge in hydrological modelling. As the time scale of our simulations corresponds to the event scale, the reviewer correctly assumes hydrological forecasts to be a potential field of application. In the revised manuscript we therefore explicitly mentioned the forecasting context. In a paper in preparation, we further investigate the use of information on dominant runoff processes in a pseudo-operational context.

**7.     P8,L29: I was a bit concerned by the fact that only six events are used to make the tests. This is not sufficient to draw general conclusion. Could the authors make an additional test in their discussion section, in which a larger number of events would be tested, to see if this corroborates the detailed results on a few events? I think this would be helpful. If this is not feasible due to data or time constraints, this aspect should be better discussed. What is the feeling of the authors on possible results in other conditions/catchments?**

Reply: The focus of the paper is more on evaluating the two different modelling approaches (i.e. bottom-up and top-down) rather than the quality of the rainfall datasets. We therefore believe that the number and type of events considered is representative and justifies our findings, at least for the catchments investigated. Furthermore, we explicitly refer to our work as a "case study" in the title of the manuscript, and nowhere in the manuscript do we claim to draw general conclusions.

Unfortunately, we have to renounce from generalising our findings, as the first author recently changed job and country, which complicates the execution of additional tests on a larger number of events. In the manuscript, however, we already speculate on how the results would look in other catchments. In those consisting of subcatchments with contrasting reactions to heavy rainfall, for instance, we believe that more complex mapping approaches would outperform the simpler ones (P16, L5-7 of the original manuscript or P17, L12-13 of the revised manuscript). We would prefer to abstain from further extending the discussion on this aspect, as it would consist of pure speculation, not supported by the results. We encourage future research to address this topic.

**8. P9,L20: A ideal P-factor equal to 1 means that you assume that the confidence intervals built in the modelling process with parameter ensembles are 100% confidence intervals. This may be mentioned since this is not obvious. Predictive uncertainty bounds may be built to contain e.g. 80% or 90% of the observations.**

Reply: In the original method described e.g. Abbaspour et al. (2007), the P-factor is quantified by the 95% prediction uncertainty calculated at the 2.5% and 97.5% levels of the cumulative distribution of the simulated runoff. However, in our study, we exclusively used the first 10 runs of the Monte Carlo simulation (for the bottom-up approach) and the first 10 runs satisfying the parameter and process constraints (for the top-down approach). In the revised manuscript we specified that we adapted these objective functions to our scope.

**9. P10,L20, P14,L15-20 and P15,L11-14: The Trueb catchment seems to be very special. Precipitation and streamflow are partly blamed for that in the article. Would there be a way to confirm these hypotheses based on the data used?**

Reply: The Trueb catchment is located very close to the meteorological station of Napf, which is an automatic station located at 1404 m a.s.l. (http://www.meteoswiss.admin.ch/home/weather/measurement-values/measurement-values-at-meteorological-stations.html?param=temperature&station=nap, accessed on 16.01.2018). Of course, the Napf measurements can lead to an overestimation of precipitation in the neighbourhood with lower altitude. This is especially true for the datasets based on interpolation with Thiessen polygons and IDW, as no height-dependent correction was undertaken. However, it is not possible to exclude that this mountain gauge affects the CombiPrecip data too. A possible way to confirm this hypothesis would be to exclude to Napf station from the datasets, but we believe that this would not change the findings of our study.

A possible way to verify potential underestimations in the measured runoff could be that used by Scherrer AG (2012; available only in German), already mentioned in our manuscript (P15,L14). In their study, they used a bottom-up method of Scherrer and Naef (2003) for identifying the extent and distribution of DRPs in the Ilfis catchment, as well as in some of its tributaries. With regard to the Trueb catchment, they observed that, given the catchment characteristics (steep hillslopes, impermeable geology), the catchment reaction to heavy rainfall is actually stronger than what was measured by the runoff gauge there in the past.

**10. P13,L1-2: My question is: what would be the objective of such a detailed representation if one had the knowledge and computational efficiency? It is as if one would like to make a map of the catchment at the 1:1 scale. Is there any use/objective for that?!**

Reply: Better information on macropores size and distribution, for instance, would be of advantage for those models explicitly considering macropores, e.g. RoGeR (Steinbrich et al., 2016), LARSIM (Bremicker, 2000) etc. Of course, even if one had the knowledge and computational efficiency, there would still be unknown unknowns to deal with (Di Baldassarre et al., 2016).

**11. P13,L3-10: I missed a clear definition of "realism" in the article. The authors make a parallel between realism and the experimentalist expectation. I would say that the experimentalist has his own model in mind, based on his knowledge, experience, etc. Therefore, confronting the numerical model with experimentalist expectation is no more than confronting two models. Here one expects that the experimentalist better knows how the catchments behaves, but the experimentalist, like any other model, may be wrong also!**

**This issue of defining realism is crucial, but I find that it should be based on objective measures. I think this would need a more in-depth discussion.**

Reply: The reviewer criticises our definition of model realism by claiming that the experimentalist's expectation, i.e. the perceptual model (Beven, 2012), may be wrong, too. He suggests using objective measures instead. However, these have problems, too. (i) They are calculated on the basis of measurements, which may be uncertain (e.g. Westerberg et al., 2011), and (ii) they are calculable only at gouging stations, which are not available everywhere. There is, therefore, a need for spatially distributed objective functions for verifying our models anywhere within the catchment. One could think of using e.g. remote sensing data or isotopes, but these are linked with uncertainty too. Since uncertainty is everywhere, why should we keep evaluating our models only against measurements, which are uncertain too?

Furthermore, the probability that the experimentalist may be wrong is lower than the chance that a model is wrong, because the perceptual model can be more complex than the mathematical one, which is composed by equations (Beven, 2000, 2012).

In a previous study, we tried to apply spatially distributed similarity measures for evaluating automatically derived process maps (Antonetti et al., 2016). As reference, we used a process map derived manually with the method of Scherrer and Naef (2003). Such a manually derived map is derived based on sprinkling experiments, soil investigations and information about the site, but is still a "model" and may therefore be wrong, too. Of course, we do not know what the true mapping is, but we can assert with a high degree of acceptance, that manually derived process maps correspond to the product of highest quality (closest to reality) a modeller can get from an experimentalist. The same is valid here, as the perceptual model of the experimentalist will be the closest to reality and, therefore, it will be worth it to use it as reference.

**12.     P15,L10-11: I found this sentence unclear.**

Reply: In the revised manuscript, we split the sentence in two parts as follows: " Results from the ANOVA also showed a considerable increase in uncertainty with decreasing size of the sub-catchments. This was also found by Hellebrand et al. (2011), who attributed it to a wrong choice of the calibration catchment."

**13.     P16,L4-7: I do agree that this is a strong limitation of this study, which prevents from reaching more general conclusions. I think the authors should try to show tests at least on a larger number of events on their catchments if possible (see comment above). I also think that a short sentence should be added in the abstract to acknowledge this limitation in the conclusion of the study.**

Reply: See answer to comment 7. We believe that:

      (i) referring in the title to a "case study" and
      (ii) explicitly claiming the restricted number of events considered as a limitation of the study

should be enough to prevent the reader from drawing general conclusions based on our study.

**14.     Table 3: Write "for the bottom-up and top-down" in the caption to keep the same order as the result tables.**

Reply: Done.

**15.     Table 4: "THY" in the caption and "THI" in the table, to be harmonized. Maybe "No. of ground rain stations…".**

Reply: Done.

**16.    Fig. 3: Maybe use the same order of maps as in table 1.**

Reply: Done.

**17.    Fig. 4: If possible, put "-1" as exponent in the unit.**

Reply: Done.

**18.    Fig. 8: Even if there are negative values for the Trueb catchment, I think that the median values of distributions not appearing on the graphs could be written in the graphs (with small arrows showing that this is down the bounds).**

Reply: Agreed. In the revised figure we put the median values in brackets. Instead of using arrows we preferred to explain the meaning of the values in the caption.

**19.    Fig. 9: "the four study catchments"**

Reply: Corrected.

**PS: I didn't find in the main text references to Figures 2 and 3.**

Reply: In the revised manuscript we added these missing references.

Bremicker, M.: Das Wasserhaushaltsmodell LARSIM - Modellgrundlagen und Anwendungsbeispiele. Freiburger Schriften zur Hydrologie, Band 11. Institut für Hydrologie der Universität Freiburg, 2000.

[revised manuscript text omitted]

---

## Author Response (AR2)

**Dear Authors,**

**The revised version gives responses to the main recommendations of both reviewers.**

**However, some minor recommendations are formulated by Reviewer #2. Please also introduce more discussions in the paper on the points you rejected following the initial recommendations of Reviewer #1.**

**Kind regards,**

**Roger Moussa**

Dear editor,

Thank you very much for your decision concerning our revised manuscript.

The minor recommendations arose from the second round of reviews, as well as the initial recommendations of anonymous referee nr.1, concern the definition of "expert knowledge" and the mention of other types of knowledge in the paper.

In the new revised version of our manuscript, therefore, we embraced the request of anonymous referee nr.1 by extending the definition of "expert knowledge" and putting it at the beginning of the introductory section. As, in our opinion, the reference to other types of knowledge at the beginning of the introduction would just distract the reader from the real topic of the paper, we decided to acknowledge the existence of other types of knowledge (e.g. citizen science) in the discussion part, namely in the section about the limitation of the study.

By doing so, we believe the revised manuscript improved in quality and we are therefore looking forward for your decision concerning its publication on Hydrology and Earth System Sciences.

Best regards,

Manuel Antonetti

Massimiliano Zappa

[revised manuscript text omitted]

---

## Author Response (AR3)

**Dear Authors,**

**The first version of the revised version gives responses to the main recommendations of both reviewers. Hoever, in the second revised version I didn't find:**

**- corrections and responses to the recommendations of Reviewer #2 (see below).**

**- more discussions in the paper on the points you rejected following the initial recommendations of Reviewer #1.**

**Kind regards,**

**Roger Moussa**

Dear Editor,

Thank you for your response concerning our study on "How can expert knowledge increase the realism of conceptual hydrological models? A case study based on the concept of dominant runoff process in the Swiss Pre-Alp".

Apparently, there has been a misunderstanding in the submission of our second revised version of our manuscript. As the second revised version was going to be reviewed exclusively by the editor, we summarised our corrections and merged our responses to the recommendations of both editor and Reviewer #2 in a single document. We apologise for this inconvenience and provide, this time, a separated reply to the report of Reviewer #2.

With regard to your second request concerning more discussion in the paper on the points we rejected from the initial recommendation of Reviewer 1, we have to admit that we did not fully understand which points should we further deepen. For clarification purposes, we listed in the following the aspects we rejected from the initial recommendations of reviewer 1, we mentioned why we rejected them, and we indicated how these points were handled within the manuscript:

1. Comparison of expert knowledge with other kinds of knowledge: We initially rejected this point by claiming that expert knowledge should not be contrasted to other kinds of knowledge, given that the term "expert", in this context, should be understood as "relating to a person who has knowledge in a particular field", which is a citation from the WordReference Random House Learner's Dictionary of American English © (2017). After the second round of review, however, we contemplated the juxtaposition of "expert" to "general" knowledge, which is the basis of the emerging field of citizen sciences. We therefore mentioned this other kind of knowledge in the revised manuscript (P17, L3-8). As, in our opinion, the reference to other types of knowledge at the beginning of the introduction would just distract the reader from the real topic of the paper, we decided to discuss the existence of other types of knowledge in the section about the limitation of the study.

2. Categorising expert knowledge: Reviewer 1 complained a lack of examples of the different types of expert knowledge. We replied that rather than types we refer to *ways* of using expert knowledge during the three steps of the process-based modelling step (i.e. process mapping, process parameterisation, and parameter allocation). The whole manuscript in fact describes, compares and evaluates different ways of implementation of expert knowledge.

3. Model realism: Here we rejected the reviewer's definition of the expression "improving the realism", which is, in our opinion, not only a question of matching performance criteria, but also an attempt of letting the model behave according to the hydrologist perception of how

it should behave. An entire paragraph is dedicated to the need of finding an agreement on what is meant with this expression (P14, L3-16), and the topic is recall on page 15, L32-33.

We believe we addressed each above-mentioned point in a balanced way in the manuscript and cannot identify aspects that need more discussion. We would therefore really appreciate a more specific definition of which points we could elaborate on.

Best regards,

Manuel Antonetti

Massimiliano Zappa

**Review of "How can expert knowledge increase the realism of conceptual hydrological models? A case study in the Swiss Pre-Alps" HESS-2017-322 by Antonetti and Zappa**

**The authors have provided a detailed response to the issues raised in the original review.**

Reply: We want to thank Reviewer nr. 2 for his/her review of our revised manuscript and are happy to acknowledge that he/she was satisfied by our response to the issues raised during the previous review.

**The only aspect that remained confusing and in my opinion warrants better clarification is with respect to clarifying "expert knowledge" in hydrology, and contrasting it to other types of knowledge (potentially) available to a hydrologist. The response says that: "In the revised manuscript we will therefore add a definition of 'expert knowledge' at the beginning of the introductory section and will distinguish in a clearer way between (i) expert knowledge per se and (ii) (top-down or bottom-up) strategies for the use of expert knowledge in conceptual models". The corresponding change I am seeing is the addition of "(i.e. someone's acquaintance as a result of study and experience)" in the first sentence of the introduction, on Line 5. To me as a hydrologist this very brief definition is only partially helpful. Under this definition, what other way is there to gain knowledge other than through "study and experience"?**

Reply: In the new revised version of our manuscript we embraced the request of anonymous reviewer 2 by extending the definition of "expert knowledge" and putting it at the beginning of the introductory section.

Another way to gain knowledge other than through "study and experience" could be represented, for instance, by citizen science, which is based on the "general" knowledge provided by amateur scientists. This other type of knowledge was, however, not contemplated within our study. As, in our opinion, the reference to other types of knowledge at the beginning of the introduction would just distract the reader from the real topic of the paper, we decided to discuss the existence of other types of knowledge (e.g. from citizen science) in the discussion part, namely in the section about the limitation of the study.

**If the manuscript singles out "expert" knowledge as its central study topic, it should provide a clear definition in the context of hydrological sciences and some examples of what IS and what IS NOT expert knowledge. The author's responses that "As acquaintance is uncountable by definition, information to be considered as 'expert knowledge' cannot be listed" and "it [expert knowledge] should not be contrasted to other kinds of knowledge" do not seem convincing to me, at least not in an international research journal in hydrology. For example, if the authors can contrast "modellers" vs "experimentalists" - despite these not being truly sharp distinctions, what is so special about "expert" knowledge that defies contrasting it to other type of knowledge? Just because it has not been done in previous cited publications is not really a scientific reason to not do it here.**

**Perhaps there is no clear definition and in fact all knowledge is "expert knowledge" - in which case this might be noted in the paper (intro/discussion).**

**I therefore recommend a revision of the manuscript to clarify this issue.**

**This request is not intended as a nit-picking or otherwise "tricky" comment - just a request as a reader/reviewer to clarify one of the central concepts in this work.**

Reply: At the beginning of the review process we were against a juxtaposition of expert knowledge to other types of knowledge. We were therefore close to the reviewer's statement that, perhaps, "all knowledge is expert knowledge". Eventually, after long thoughts, we decided to embrace the juxtaposition of expert (i.e. "linked with study and experience") to general knowledge, as a tribute to the emerging field of citizen science. However, this does not change the focus of our manuscript, which remains on the different ways of applying expert knowledge in the modelling chain based on the concept of dominant runoff process. Therefore, the discussion on this other kind of knowledge was relegated to the section about the limitation of the study.

In our replies to the reviewers we never ever used a hasty, non-scientific explanation like "since it has not been done in previous publications we are not doing it here", nor intended to insinuate something like this at all! If the reviewer refers to the list of publications, where the expression "expert knowledge" was already used, it served exclusively to support our decision to keep the expression as is for consistency.

[revised manuscript text omitted]